# GRAPH NEURAL MODELING OF NETWORK FLOWS

## ABSTRACT

Network flow problems, which involve distributing traffic such that the underlying infrastructure is used effectively, are ubiquitous in transportation and logistics. Among them, the general Multi-Commodity Network Flow (MCNF) problem concerns the distribution of multiple flows of different sizes between several sources and sinks, while achieving effective utilization of the links. Due to the appeal of data-driven optimization, these problems have increasingly been approached using graph learning methods. In this paper, we propose a novel graph learning architecture for network flow problems called Per-Edge Weights (PEW). This method builds on a Graph Attention Network and uses distinctly parametrized message functions along each link. We extensively evaluate the proposed solution through an Internet flow routing case study using 17 Service Provider topologies and 2 routing schemes. We show that PEW yields substantial gains over architectures whose global message function constrains the routing unnecessarily. We also find that an MLP is competitive with other standard architectures. Furthermore, we analyze the relationship between graph structure and predictive performance for data-driven routing of flows, an aspect that has not been considered by existing work in the area.

## 1 INTRODUCTION

Flow routing represents a fundamental problem that captures a variety of optimization scenarios that arise in real-world networks (Ahuja, 1993, Chapter 17). One classic example is the maximum flow problem, which seeks to find the best (in terms of maximum capacity) path between a source node and a sink node. The more general Multi-Commodity Network Flow problem allows for multiple flows of different sizes between several sources and sinks that share the same distribution network. It is able to formalize the distribution of packets in a computer network, of goods in a logistics network, or cars in a rail network (Hu, 1963). We illustrate MCNF problems in Figure 1.

For maximum flow problems, efficient algorithms have been developed (Cormen et al., 2022, Chapter 26), including a recent near-linear time approach (Chen et al., 2022). For the more complex MCNF problems, Linear Programming solutions can be leveraged in order to compute, in polynomial time, the optimal routes given knowledge of pairwise demands between the nodes in the graph (Fortz & Thorup, 2000; Tardos, 1986). At the other end of the spectrum, oblivious routing methods derive routing strategies with partial or no knowledge of traffic demands, optimizing for "worst-case" performance (Räcke, 2008).

As recognized by existing works, *a priori* knowledge of the full demand matrix is an unrealistic assumption, as loads in real systems continuously change (Feldmann et al., 2001). Instead, ML techniques may enable a middle ground (Valadarsky et al., 2017): learning a model trained on past loads that can perform well in a variety of traffic scenarios, without requiring a disruptive redeployment of the routing strategy (Fortz & Thorup, 2002). Hence, developing an effective learning representation is fundamental to the application of ML in flow routing scenarios.

From a more practical point of view, this shift towards data-driven approaches is illustrated by the concepts of data-driven computer networking (Jiang et al., 2017) and self-driving networks (Feamster & Rexford, 2017). Early works in this area were based on MLP architectures (Valadarsky et al., 2017; Reis et al., 2019). More recently, models purposely designed to operate on graphs, including variants of the expressive Message Passing Neural Networks (Rusek et al., 2019; Almasan et al., 2021) and Graph Nets (Battaglia et al., 2018), have been adopted.

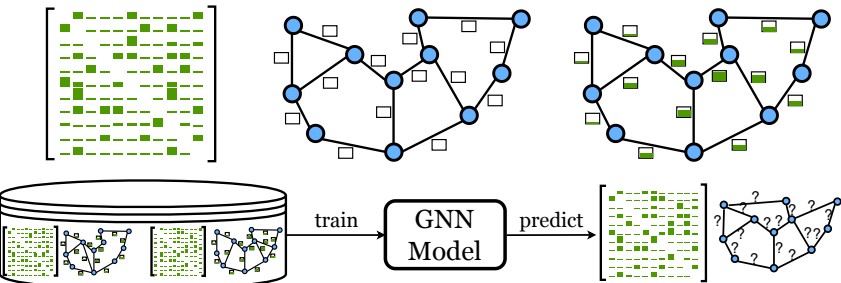

Figure 1: **Top.** An illustration of the Multi-Commodity Network Flow family of problems. The requirements of the routing problem are defined using a matrix that specifies the total amount of traffic that has to be routed between each pair of nodes in a graph. We are also given a graph topology in which links are equipped with capacities. All flows have an entry and exit node and share the same underlying transportation infrastructure. Under a particular routing scheme, such as shortest path routing, the links are loaded by the total amount of traffic passing over them. **Bottom.** A model is trained using a dataset of the link utilizations for certain demand matrices and graph topologies, and is then used to predict the Maximum Link Utilization for an unseen demand matrix.

Despite the promise of graph learning, current works nevertheless adopt schemes that aggregate messages along neighboring edges using the same message functions. In the context of routing flows, this constrains the model unnecessarily. Instead, we argue that nodes should be able to weight flows along each link separately, so that each node may independently update its state given incoming and outgoing traffic, leading to better *algorithmic alignment* (Xu et al., 2020) between the computational mechanism of the GNN and the task. We illustrate this in Figure 2.

Furthermore, the ways in which prior works encode the demands as node features varies between the full demand matrix (Valadarsky et al., 2017; Zhang et al., 2020) and a node-wise summation (Hope & Yoneki, 2021), and it is unclear when either is beneficial. Besides the learning representation aspects, existing approaches in this area are trained using very few graph topologies (typically 1 or 2) of small sizes (typically below 20 nodes). This makes it difficult to assess the gain that graph learning solutions bring over vanilla architectures such as the MLP. Additionally, a critical point that has not been considered is the impact of the underlying graph topology on the effectiveness of the learning process. To address these shortcomings, we make contributions along the following axes:

- **Learning representations for data-driven flow routing.** We propose a novel mechanism for aggregating messages along each link with a different parametrization, which we refer to as *Per-Edge Weights (PEW)*. We propose an instantiation that extends the GAT (Veličković et al., 2018) via a construction akin to the RGAT (Busbridge et al., 2019). Despite its simplicity, we show that this mechanism yields substantial predictive gains over architectures that use the same message function for all neighbors. We also find that PEW can exploit the complete demand matrix as node features, while the GAT performs better with the lossy node-wise sum used in prior work.

- **Rigorous and systematic evaluation.** Whereas existing works test on few, small-scale topologies, we evaluate the proposed method and 4 baselines on 17 real-world Internet Service Provider topologies and 2 routing schemes in the context of a case study in computer networks, yielding 81600 independent model training runs. Perhaps surprisingly, we find that a well-tuned MLP is competitive with other GNN architectures when given an equal hyperparameter and training budget.

- **Understanding the impact of topology.** The range of experiments we carry out allows us to establish that a strong link exists between topology and the difficulty of the prediction task, which is consistent across routing schemes. Generally, the predictive performance decreases with the size of the graph and increases with heterogeneity in the local node and edge properties. Moreover, we find that, when graph structure varies through the presence of different subsets of nodes, the predictive performance of GNNs increases compared to structure-agnostic methods, such as MLP.

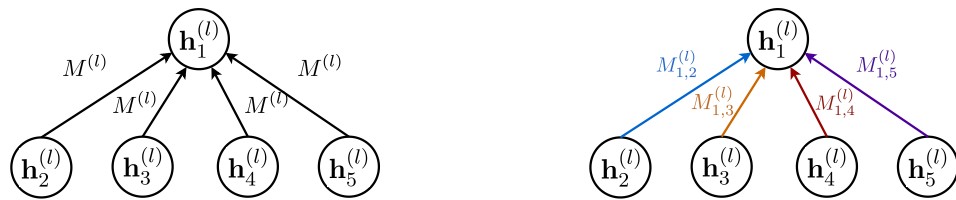

Figure 2: **Left.** An illustration of the MPNN used in previous flow routing works, which uses the same message function $M^{(l)}$ for aggregating neighbor messages. **Right.** An illustration of our proposed Per-Edge Weights (PEW), which uses uniquely parametrized per-edge message functions.

## 2 RELATED WORK

**Neural networks operating on graphs.**   Much effort has been devoted in recent years to developing neural network architectures operating on graphs. Several approaches, such as Graph Convolutional Networks (GCNs), use convolutional filters based on the graph spectrum, which can be approximated efficiently (Defferrard et al., 2016; Kipf & Welling, 2017). An alternative line of work is based on message passing on graphs (Sperduti & Starita, 1997; Scarselli et al., 2009) as a means of deriving vectorial embeddings. Both Message Passing Neural Networks (MPNNs) (Gilmer et al., 2017) and Graph Networks (Battaglia et al., 2018) are attempts to unify related methods in this space, abstracting the commonalities of existing approaches with a set of primitive functions.

Expressivity is another major concern in the design of this class of architectures. Notably, Gated Graph Neural Networks (GG-NNs) (Li et al., 2017) add gating mechanisms and support for different relation types, as well as removing the need to run the message propagation to convergence. Graph Attention Networks (GATs) (Veličković et al., 2018) propose the use of attention mechanisms as a way to perform flexible aggregation of neighbor features. Relational learning models for knowledge graphs, such as the RGCNs (Schlichtkrull et al., 2018) that extends the GCN architecture, use different parametrizations for edges with different types. The RGATs (Busbridge et al., 2019) follow the blueprint of RGCNs and extend the GAT approach to the relational setting. Despite the tremendous success of relational models for a variety of tasks, perhaps surprisingly, recent work shows that randomly trained relation weights may perform similarly well (Degraeve et al., 2022).

**ML for routing flows in computer networks.**   Several works have considered machine learning approaches to perform supervised learning for routing flows in computer networks. (Geyer & Carle, 2018) proposes a variant of the GG-NN and trains it to predict paths taken by conventional routing algorithms. (Rusek et al., 2019) proposes a MPNN variant and uses it to predict graph-level metrics such as delay and jitter. (Reis et al., 2019) uses an MLP representation and supervised learning to predict the full path that a flow should take through the network. Other works have considered reinforcement learning the routing protocol itself in a variety of problem formulations: (Valadarsky et al., 2017) uses an MLP and considers learning per-edge coefficients that are used with "softmin" routing. (Xu et al., 2018) proposes an MLP approach for learning traffic split ratios for a set of candidate paths.  (Zhang et al., 2020) uses a CNN to re-route a proportion of important (critical) flows. (Almasan et al., 2021) proposes a formulation that routes flows sequentially, which then become part of the state. It uses a MPNN representation. Most recently, (Hope & Yoneki, 2021) adopts the formulation in (Valadarsky et al., 2017), showing that the use of Graph Networks improves performance in one graph topology.

**Algorithmic reasoning.**   Another relevant area is algorithmic reasoning (Veličković et al., 2018; Cappart et al., 2021), which trains neural networks to execute the steps taken by classic algorithms (Cormen et al., 2022) with the goal of obtaining strong generalization on larger unseen inputs. (Georgiev & Liò, 2020) trains a MPNN to mimic the steps taken by the Ford-Fulkerson maximum flow algorithm (Cormen et al., 2022, Chapter 24). An important difference to this line of work is that in our case our model does not include any knowledge of the routing scheme, while the approaches based on algorithmic reasoning use the granular algorithm steps themselves as the supervision signal.

## 3 METHODS

### 3.1 ROUTING FORMALIZATION AND LEARNING TASK

**Flow routing formalization.** We assume the splittable-flow routing formalization proposed by Fortz & Thorup (2004). We let $G = (V, E)$ be a directed graph, with $V$ representing the set of nodes and $E$ the set of edges. We use $N = |V|$ and $m = |E|$ as shorthands, as well as $v_i$ and $e_{i,j}$ to denote specific nodes and edges, respectively. Each edge has an associated *capacity* $\kappa(e_{i,j}) \in \mathbb{R}^+$. We also define a *demand matrix* $\mathbf{D} \in \mathbb{R}^{N \times N}$ where entry $D_{src,dst}$ is the traffic that source node $src$ sends to destination $dst$. With each tuple $(src, dst, e_{i,j}) \in V \times V \times E$ we associate the quantity $f_{e_{i,j}}^{(src,dst)} \geq 0$, which specifies the amount of traffic flow from $src$ to $dst$ that goes over the edge $e_{i,j}$. The *load* of edge $e_{i,j}$, $\text{load}(e_{i,j})$, is the total traffic flow traversing it, i.e., $\text{load}(e_{i,j}) = \sum_{(src,dst) \in V \times V} f_{e_{i,j}}^{(src,dst)}$. Furthermore, the quantities $f_{e_{i,j}}^{(src,dst)}$ must obey the following flow conservation constraints:

$$\sum_{e \in \delta^+(v_i)} f_e^{(src,dst)} - \sum_{e \in \delta^-(v_i)} f_e^{(src,dst)} = \begin{cases} D_{src,dst} & \text{if } v_i = src, \\ -D_{src,dst} & \text{if } v_i = dst, \\ 0 & \text{otherwise.} \end{cases} \tag{1}$$

where the sets $\delta^+(v_i), \delta^-(v_i)$ are node $v_i$'s outgoing and incoming edges, respectively. Intuitively, these constraints capture the fact that traffic sent from $src$ to $dst$ originates at the source (first clause), must be absorbed at the target (second clause), and ingress equals egress for all other nodes (final clause).

**Routing schemes.** A routing scheme $\mathscr{R}$ specifies how to distribute the traffic flows. Specifically, we consider two well-known routing schemes. The first is the *Standard Shortest Paths* (SSP) scheme in which, for a given node, the full flow quantity with destination $dst$ is sent to the neighbor on the shortest path to $dst$. The widely used *ECMP* scheme (Hopps, 2000) instead splits outgoing traffic among all the neighbors on the shortest path to $dst$ if multiple such neighbors exist.

**Prediction target.** A common way of evaluating a routing strategy $\mathscr{R}$ is *Maximum Link Utilization (MLU)*, i.e., the maximal ratio between link load and capacity. Formally, given a demand matrix $\mathbf{D}$, we denote it as $\text{MLU}(\mathbf{D}) = \max_{e_{i,j} \in E} \frac{\text{load}(e_{i,j})}{\kappa(e_{i,j})}$. This target metric has been extensively studied in prior work (Kandula et al., 2005) and is often used by ISPs to gauge when the underlying infrastructure needs to be upgraded (Guichard et al., 2005).

**Supervised learning setup.** We assume that we are provided with a dataset of traffic matrices $\mathcal{D} = \cup_k \{\mathbf{D}^{(k)}, \text{MLU}(\mathbf{D}^{(k)})\}$. Given that our model produces an approximation $\widehat{\text{MLU}}(\mathbf{D}^{(k)})$ of the true Maximum Link Utilization, the goal is to minimize the Mean Squared Error $\frac{\sum_k (\text{MLU}(\mathbf{D}^{(k)}) - \widehat{\text{MLU}}(\mathbf{D}^{(k)}))^2}{|\mathcal{D}|}$.

### 3.2 PER-EDGE WEIGHTS

We propose a simple mechanism to increase the expressivity of models for data-driven flow routing. As previously mentioned, several works in recent years have begun adopting various graph learning methods for flow routing problems such as variants of Message Passing Neural Networks (Geyer & Carle, 2018; Rusek et al., 2019; Almasan et al., 2021) or Graph Networks (Hope & Yoneki, 2021). In particular, MPNNs derive hidden features $\mathbf{h}_{v_i}^{(l)}$ for node $v_i$ in layer $l+1$ by computing messages $\mathbf{m}^{(l+1)}$ and applying updates of the form:

$$\mathbf{m}_{v_i}^{(l+1)} = \sum_{v_j \in \mathcal{N}(v_i)} M^{(l)} \left( \mathbf{h}_{v_i}^{(l)}, \mathbf{h}_{v_j}^{(l)}, \mathbf{x}_{e_{i,j}} \right)$$
$$\mathbf{h}_{v_i}^{(l+1)} = U^{(l)} \left( \mathbf{h}_{v_i}^{(l)}, \mathbf{m}_{v_i}^{(l+1)} \right) \tag{2}$$

where $\mathcal{N}(v_i)$ is the neighborhood of node $v_i$, $\mathbf{x}_{e_{i,j}}$ are features for edge $e_{i,j}$, and $M^{(l)}$ and $U^{(l)}$ are the differentiable message (sometimes also called edge) and vertex update functions in layer

$l$. Typically, $M^{(l)}$ is some form of MLP that is applied in parallel when computing the update for each node in the graph. An advantage of applying the same message function $M^{(l)}$ across the entire graph is that the number of parameters remains fixed in the size of the graph, enabling a form of combinatorial generalization (Battaglia et al., 2018). However, while this approach has been very successful in many graph learning tasks such as graph classification, we argue that it is not best suited for flow routing problems.

Instead, for this family of problems, the edges do not have uniform semantics. Each of them plays a different role when the flows are routed over the graph and, as shown in Figure 1, each will take on varying levels of load. Equivalently, from a node-centric perspective, each node should be able to decide flexibly how to distribute several flows of traffic over its neighboring edges. This intuition can be captured by using a different message function $M_{i,j}^{(l)}$ when aggregating messages received along each edge $e_{i,j}$. We call this mechanism *Per-Edge Weights*, or *PEW*. We illustrate the difference between PEW and a typical MPNN in Figure 2.

Let us formulate the PEW architecture by a similar construction to the additive self-attention, across-relation variant of RGAT (Busbridge et al., 2019). Let $\mathcal{N}[v_i]$ and $\mathcal{N}(v_i)$ denote the closed and open neighborhoods of node $v_i$. To compute the coefficients for each edge, one first needs to compute intermediate representations $\mathbf{g}_{v_i,e_{i,j}}^{(l)} = \mathbf{W}_{e_{i,j}}^{(l)} \mathbf{h}_{v_i}$ by multiplying the node features with the per-edge weight matrix $\mathbf{W}_{e_{i,j}}^{(l)}$. Subsequently, the "query" and "key" representations are defined as below, where $\mathbf{Q}_{e_{i,j}}^{(l)}$ and $\mathbf{K}_{e_{i,j}}^{(l)}$ represent per-edge query and key kernels respectively:

$$\mathbf{q}_{v_i,e_{i,j}}^{(l)} = \mathbf{g}_{v_i,e_{i,j}}^{(l)} \cdot \mathbf{Q}_{e_{i,j}}^{(l)} \text{ and } \mathbf{k}_{v_i,e_{i,j}}^{(l)} = \mathbf{g}_{v_i,e_{i,j}}^{(l)} \cdot \mathbf{K}_{e_{i,j}}^{(l)}. \tag{3}$$

Then, the attention coefficients $\zeta_{e_{i,j}}^{(l)}$ are computed according to:

$$\zeta_{e_{i,j}}^{(l)} = \frac{\exp\left(\text{LeakyReLU}\left(\mathbf{q}_{v_i,e_{i,j}}^{(l)} + \mathbf{k}_{v_j,e_{i,j}}^{(l)} + \mathbf{W}_1^{(l)}\mathbf{x}_{e_{i,j}}\right)\right)}{\sum_{v_k \in \mathcal{N}[v_i]} \exp\left(\text{LeakyReLU}\left(\mathbf{q}_{v_i,e_{i,k}}^{(l)} + \mathbf{k}_{v_k,e_{i,k}}^{(l)} + \mathbf{W}_1^{(l)}\mathbf{x}_{e_{i,k}}\right)\right)}, \tag{4}$$

Finally, the embeddings are computed as:

$$\mathbf{h}_{v_i}^{(l+1)} = \text{ReLU}\left(\sum_{v_j \in \mathcal{N}(v_i)} \zeta_{e_{i,j}}^{(l)} \mathbf{g}_{v_j,e_{i,j}}^{(l)}\right). \tag{5}$$

## 4 EVALUATION PROTOCOL

This section describes the experimental setup we use for our evaluation. We focus on a case study on routing flows in computer networks to demonstrate its effectiveness in real-world scenarios, which can be considered representative of a variety of settings in which we wish to predict characteristics of a routing scheme from an underlying network topology and a set of observed demand matrices.

**Model architectures.** We compare PEW with three widely used graph learning architectures: the GAT (Veličković et al., 2018), GCN (Kipf & Welling, 2017), and GraphSAGE (Hamilton et al., 2017). We also compare against a standard MLP architecture made up of fully-connected layers followed by ReLU activations. The features provided as input to the five methods are the same: for the GNN methods, the node features are the demands $\mathbf{D}$ in accordance with the demand input representations defined later in this section, while the edge features are the capacities $\kappa$, and the adjacency matrix $\mathbf{A}$ governs the message passing. For GCN and GraphSAGE, which do not support edge features, we include the mean edge capacity as a node feature. For the MLP, we unroll and concatenate the demand input representation derived from $\mathbf{D}$, the adjacency matrix $\mathbf{A}$, and all edge capacities $\kappa$ in the input layer. We note that other non-ML baselines, such as Linear Programming, are not directly applicable for this task: while they can be used to derive a routing strategy, in this chapter the goal is to predict a property of an existing routing strategy (SSP or ECMP, as defined in Section 3.1).

**Traffic generation.** In order to generate synthetic flows of traffic, we use the "gravity" approach proposed by Roughan (2005). Akin to Newton's law of universal gravitation, the traffic $D_{i,j}$ between nodes $v_i$ and $v_j$ is proportional to the amount of traffic, $D_i^{\text{in}}$, that enters the network via $v_i$ and $D_j^{\text{out}}$, the amount that exits the network at $v_j$. The values $D_i^{\text{in}}$ and $D_j^{\text{out}}$ are random variables that are identically and independently distributed according to an exponential distribution. Despite its simplicity in terms of number of parameters, this approach has been shown to synthesize traffic matrices that correspond closely to those observed in real-world networks (Roughan, 2005; Hartert et al., 2015). We additionally apply a rescaling of the volume by the MLU (defined in Section 3.1) under the LP solution of the MCNF formulation, as recommended in the networking literature (Haddadi & Bonaventure, 2013; Gvozdiev et al., 2018).

**Network topologies.** We consider real-world network topologies that are part of the Repetita and Internet Topology Zoo repositories (Gay et al., 2017; Knight et al., 2011). In case there are multiple snapshots of the same network topology, we only use the most recent so as not to bias the results towards these graphs. We limit the size of the considered topologies to between $[20, 100]$ nodes, which we note is still substantially larger than topologies used for training in prior work on ML for routing flows. Furthermore, we only consider heterogeneous topologies with at least two different link capacities. Given the traffic model above, for some topologies the MLU dependent variable is nearly always identical regardless of the demand matrix, making it trivial to devise a good predictor. Out of the 39 resulting topologies, we filter out those for which the minimum MLU is equal to the 90th percentile MLU over 100 demand matrices, leaving 17 unique topologies. The properties of these topologies are summarized in the Appendix. For the experiments involving topology variations, they are generated as follows: a number of nodes to be removed from the graph is chosen uniformly at random in the range $[1, \frac{N}{5}]$, subject to the constraint that the graph does not become disconnected. Demand matrices are generated starting from this modified topology.

**Datasets.** The datasets $\mathcal{D}^{\text{train}}, \mathcal{D}^{\text{validate}}, \mathcal{D}^{\text{test}}$ of demand matrices are disjoint and contain $10^3$ demand matrices each. Both the demands and capacities are standardized by dividing them by the maximum value across the union of the datasets. As shown in the Appendix, the datasets for the smallest topology contain $1.2 * 10^6$ flows, while the datasets for the largest topology consist of $2 * 10^7$ flows.

**Demand input representation.** We also consider two different demand input representations that appear in prior work, which we term *raw* and *sum*. In the former, the feature vector $\mathbf{x}_{v_i}^{\text{raw}} \in \mathbb{R}^{2N}$ for node $v_i$ is $[D_{1,i}, \ldots, D_{N,i}, D_{i,1}, \ldots, D_{i,N}]$, which corresponds to the concatenated outgoing and incoming demands, respectively. The latter is an aggregated version $\mathbf{x}_{v_i}^{\text{sum}} \in \mathbb{R}^2$ equal to $[\sum_j D_{i,j}, \sum_i D_{j,i}]$, i.e., it contains the summed demands.

**Training and evaluation protocol.** Training and evaluation are performed separately for each graph topology and routing scheme. All methods are given an equal grid search budget of 12 hyperparameter configurations whose values are provided in the Appendix. To compute means and confidence intervals, we repeat training and evaluation across 10 different random seeds. Training is done by mini-batch SGD using the Adam optimizer (Kingma & Ba, 2015) and proceeds for 3000 epochs with a batch size of 16. We perform early stopping if the validation performance does not improve after 1500 epochs, also referred to as "patience" in other graph learning works (Veličković et al., 2018; Errica et al., 2020). Since the absolute value of the MLUs varies significantly in datapoints generated for different topologies, we apply a normalization when reporting results such that they are comparable. Namely, the MSE of the predictors is normalized by the MSE of a simple baseline that outputs the average MLU for all DMs in the provided dataset. We refer to this as Normalized MSE (NMSE).

**Scale of experiments.** Given the range of considered graph learning architectures, hyperparameter configurations, network topologies and routing models, to the best of our knowledge, our work represents the most extensive suite of benchmarks on graph learning for the MCNF problem to date. The primary experiments consist of 20400 independent model training runs, while the entirety of our experiments comprise 81600 runs. We believe that this systematic experimental procedure and evaluation represents in itself a significant contribution of our work and, akin to (Errica et al., 2020) for graph classification, it can serve as a foundation for members of the graph learning community working on MCNF scenarios to build upon.

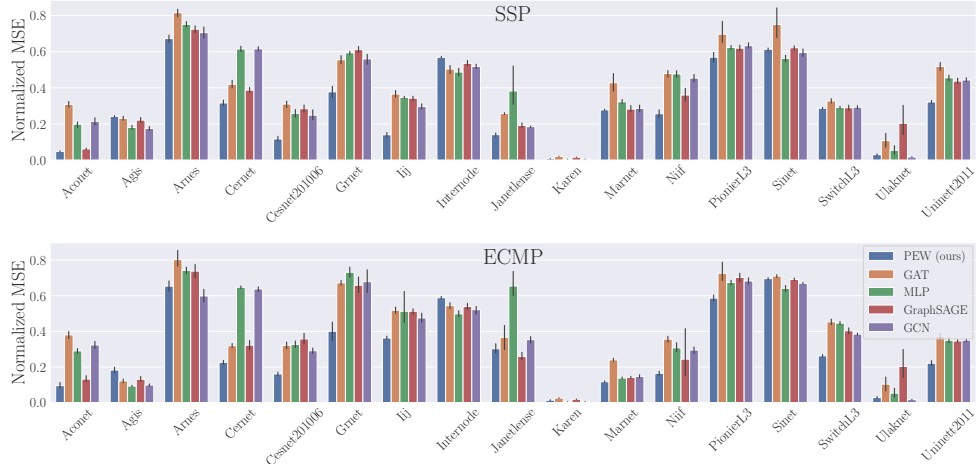

Figure 3: Normalized MSE obtained by the predictors on different topologies for the SSP (top) and ECMP (bottom) routing schemes. Lower values are better. PEW improves over vanilla GAT substantially and performs best out of all architectures. An MLP is competitive with the other GNNs.

## 5 EVALUATION RESULTS

**Benefits of PEW for flow routing.** The primary results are shown in Figure 3, in which we compare the normalized MSE obtained by the 5 architectures on the 17 topologies. The two rows correspond to the SSP and ECMP schemes respectively. Learning curves for the best-performing hyperparameter configurations are presented in the Appendix. We find that PEW improves the predictive performance over a vanilla GAT in nearly all ($88\%$) of the settings tested, and that it performs the best out of all predictors in $64.7\%$ of cases. Hence, this highlights the importance of parametrizing links differently, suggesting that it is an effective inductive bias for this family of problems. Interestingly, the MLP performs better than GAT in 80% of the considered cases, and is competitive with GCN and GraphSAGE. This echoes findings in other graph learning works (Errica et al., 2020), i.e., the fact that a well-tuned MLP can be competitive against GNN architectures and even outperform them. Furthermore, both the relative differences between predictors and their absolute normalized MSEs are fairly consistent across the different topologies.

**Varying graph structure.** Next, we investigate the impact of variations in topology on predictive performance. In this experiment, the sole difference wrt. the setup described above is that the datasets contain $10^3$ demand matrices that are instead distributed on $25$ variations in topology of the original graph (i.e., we have $40$ DMs per variation making up each dataset). To evaluate the methods, we use two ranking metrics: the Win Rate (WR) is the percentage of topologies for which the method obtains the lowest NMSE, and the Mean Reciprocal Rank (MRR) is the arithmetic average of the complements of the ranks of the three predictors. For both metrics, higher values are better. Results are shown in Table 1. PEW remains the best architecture and manifests a decrease in MRR for SSP and a gain for ECMP. We also find that the relative performance of the GCN increases while that of the MLP decreases when varying subsets of the nodes in the original graph are present. This suggests that GNN-based approaches are more resilient to changes in graph structure (e.g., nodes joining and leaving the network), a commonly observed phenomenon in practice.

**Best demand input representation.** To compare the two demand input representations, we additionally train the model architectures on subsets of $5\%, 10\%, 25\%$ and $50\%$ of the datasets. Recall that the *raw* representation contains the full demand matrix while the *sum* representation is a lossy aggregation of the same information. The latter may nevertheless help to avoid overfitting. Furthermore, given that the distribution of the demands is exponential, the largest flows will dominate the values of the features. Results are shown in Figure 4. The $x$-axis indicates the number of demand matrices used for training and evaluation, while the $y$-axis displays the difference in normalized MSE between the *raw* and *sum* representations, averaged across all topologies. As marked in the figure, $y > 0$ means that the raw representation performs better, while the reverse is true for $y < 0$.

Table 1: Mean Reciprocal Rank and Win Rates for the different predictors. PEW maintains the overall best performance. The relative performance of the MLP decreases when the graph structure varies by means of different subsets of nodes being present and generating demands.

| | | PEW (ours) | | GAT | | MLP | | GraphSAGE | | GCN | |
|---|---|---|---|---|---|---|---|---|---|---|---|
| $\mathscr{R}$ | Metric | Original $G$ | Variations | Original $G$ | Variations | Original $G$ | Variations | Original $G$ | Variations | Original $G$ | Variations |
| SSP | MRR ↑ | **0.798** | **0.747** | 0.252 | 0.240 | 0.419 | 0.396 | 0.367 | 0.349 | 0.448 | 0.551 |
| | WR ↑ | **70.588** | **58.824** | 0.000 | 0.000 | 17.647 | 11.765 | 0.000 | 5.882 | 11.765 | 23.529 |
| ECMP | MRR ↑ | **0.734** | **0.755** | 0.250 | 0.254 | 0.462 | 0.413 | 0.381 | 0.338 | 0.456 | 0.524 |
| | WR ↑ | **58.824** | **58.824** | 0.000 | 0.000 | 23.529 | 11.765 | 5.882 | 5.882 | 11.765 | 23.529 |

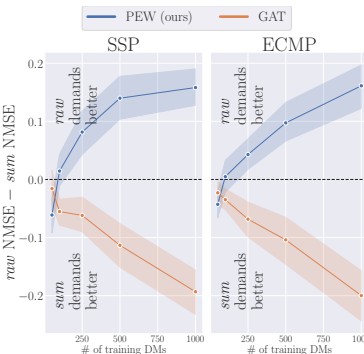

Figure 4: Difference in normalized MSE between the *raw* and *sum* demand input representations as a function of the number of training datapoints for PEW and GAT for the SSP (left) and ECMP routing schemes (right). As the dataset size increases, PEW is able to exploit the granular demand information, while GAT performs better with a lossy aggregation of the demand information.

With very few datapoints, the two input representations yield similar errors for both PEW and GAT. Beyond this, two interesting trends emerge: as the number of datapoints increases, PEW performs better with the raw demands, while the vanilla GAT performs better with the lossy representation. This suggests that, while the PEW model is able to exploit the granular information contained in the raw demands, they instead cause the standard GAT to overfit and obtain worse generalization performance.

**Impact of topology.** Our final set of experiments examines the relationship between the topological characteristics of graphs and the relative performance of our proposed model architecture. The six properties that we examine are defined as follows, noting that the first three are global properties while the final three measure the variance in local node and edge properties:

- **Number of nodes**: the cardinality $N$ of the node set $V$;

- **Diameter**: max. length among pairwise shortest paths;

- **Edge density**: the ratio of links to nodes $\frac{m}{N}$;

- **Capacity variance**: the variance in the normalized capacities $\kappa(e_{i,j})$;

- **Degree variance**: the variance in $\frac{\deg(v_i)}{N}$;

- **Weighted betweenness variance**: the variance in a weighted version of betweenness centrality (Brandes, 2001) measuring the fraction of all-pairs shortest paths passing through each node.

The results of this analysis are shown in Figure 5. As previously, the normalized MSE of the PEW model is shown on the $y$-axis, while the $x$-axis measures properties of the graphs. Each datapoint represents one of the 17 topologies. We find that topological characteristics do not fully determine model performance but, nevertheless, it is possible to make a series of observations related to them. Generally, the performance of the method decreases as the size of the graph grows in number of nodes, diameter, and edge density (metrics that are themselves correlated). This result can be explained by the fact that our experimental protocol relies on a fixed number of demand matrices, which represent a smaller sample of the distribution of demand matrices as the graph increases in size. Hence, this can lead to a model with worse generalization from the training to the test phase, despite the larger parameter count. On the other hand, the performance of the method typically improves with increasing heterogeneity in node and link-level properties (namely, variance in the capacities and degree / weighted betweenness centralities). The relationship between the NMSE and some properties (e.g., weighted betweenness) may be non-linear. Additional results that relate topological characteristics to the *percentage changes* in NMSE from the other architectures to PEW are presented in the Appendix. These analyses further corroborate the findings concerning the relationship between the predictive performance of PEW and graph structure.

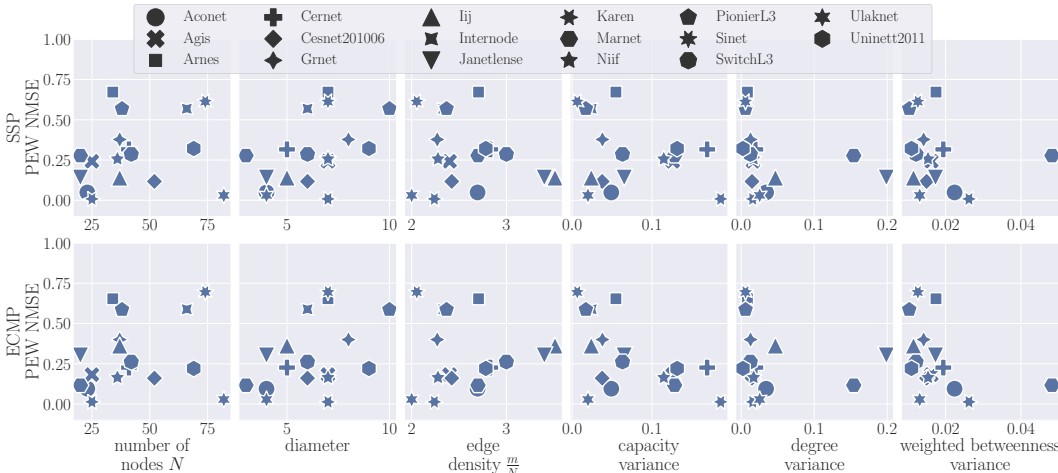

Figure 5: Impact of topological characteristics on the predictive performance of PEW. Performance degrades as the graph size increases (first 3 columns), but improves with higher levels of heterogeneity of the graph structure (last 3 columns).

## 6  CONCLUSION, LIMITATIONS, AND FUTURE WORK

**Summary.** In this work, we have addressed the problem of data-driven routing of flows across a graph, which has several applications of practical relevance in areas as diverse as logistics and computer networking. We have proposed Per-Edge Weights (PEW), an effective model architecture for predicting link loads in a network based on historical observations, given a demand matrix and a routing strategy. The novelty of our approach resides in the use of weight parametrizations for aggregating messages that are unique for each edge of the graph. In a rigorous and systematic evaluation comprising 81600 training runs, we have demonstrated that PEW improves predictive performance over standard graph learning and MLP approaches. Furthermore, we have shown that PEW is able to exploit the full demand matrix, unlike the standard GAT, for which a lossy aggregation of features is preferable. Our findings also highlight the importance of topology for data-driven routing. Given the same number of historical observations, performance typically decreases when the graph grows in size, but increases with higher levels of heterogeneity of local properties.

**Limitations.** A possible disadvantage of PEW is that the number of parameters grows linearly with the edge count. However, since the same amount of computations are performed, there is no increase in runtime compared to the GAT. Additionally, given the relatively small scale of ISP backbone networks (several hundreds of nodes), in practice, the impact on memory usage has not been significant in our experiments. The largest PEW model, used for the Uninett2011 graph, has approximately $8 * 10^5$ parameters. If required, approaches for reducing the parameter count, such as the basis and block-diagonal decompositions proposed by Schlichtkrull et al. (2018), have already been validated for significantly larger-scale relational graphs. Other routing-specific options that may be investigated in future work could be the "clustering" of the edges depending on the structural roles that they play (such as peripheral or core links) or the use of differently parametrized neighborhoods for the regions of the graph, which may perform well if a significant proportion of the traffic is local. Furthermore, a key assumption behind PEW is that node identities are known, so that when topologies vary, the mapping to a particular weight parametrization is kept consistent. This is a suitable assumption for a variety of real-world networks, such as the considered ISP backbone networks, which are characterized by infrequent upgrades. However, performance may degrade in highly dynamic networks, where the timescale of the structural changes is substantially lower than the time needed in order for systems making use of such a predictive model to adapt.

**Future work.** While this paper has focused on learning the properties of *existing* routing protocols, in future work we aim to investigate learning *new* routing protocols given the proposed learning representation and broader insights in this problem space that we have obtained.

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

## A  IMPLEMENTATION AND RUNTIME DETAILS

**Implementation.** Please consult the `README.md` file for instructions on how to run the provided code. In the case of acceptance, our implementation will be made publicly available as Docker containers together with instructions that enable reproducing (up to hardware differences) all the results reported in the paper, including tables and figures. We implement all approaches and baselines in Python using a variety of numerical and scientific computing packages (Hunter, 2007; Hagberg et al., 2008; McKinney, 2011; Paszke et al., 2019; Waskom, 2021). For implementations of the graph learning methods, we make use of PyTorch Geometric (Fey & Lenssen, 2019). Due to the relationship between the RGAT and PEW architectures, we are able to leverage the existing RGAT implementation in this library.

**Data availability.** The network topology data used in this paper is part of the Repetita suite (Gay et al., 2017) and it is publicly available at `https://github.com/svissicchio/Repetita` without any licensing restrictions. We also use the synthetic traffic generator from (Gvozdiev et al., 2018), available at `https://github.com/ngvozdiev/tm-gen`.

**Infrastructure and runtimes.** Experiments were carried out on a cluster of 8 machines, each equipped with 2 Intel Xeon E5-2630 v3 processors and 128GB RAM. On this infrastructure, all the experiments reported in this paper took approximately 35 days to complete. The training and evaluation of models were performed exclusively on CPUs.

## B HYPERPARAMETER DETAILS

Table 2: Hyperparameters used.

| | PEW (ours) | GAT | MLP | GraphSAGE | GCN |
|---|---|---|---|---|---|
| Learning rates $\alpha$ | $\{10^{-2}, 5*10^{-3}, 10^{-3}\}$ | $\{10^{-2}, 5*10^{-3}, 10^{-3}\}$ | $\{10^{-2}, 5*10^{-3}, 10^{-3}\}$ | $\{10^{-2}, 5*10^{-3}, 10^{-3}\}$ | $\{10^{-2}, 5*10^{-3}, 10^{-3}\}$ |
| Demand input representations | $\{raw, sum\}$ | $\{raw, sum\}$ | $\{raw, sum\}$ | $\{raw, sum\}$ | $\{raw, sum\}$ |
| Dimension of feature vector $\mathbf{h}$ | $\{4, 16\}$ | $\{8, 32\}$ | n/a | $\{8, 32\}$ | $\{8, 32\}$ |
| First hidden layer size | n/a | n/a | $\{64, 256\}$ *sum* / $\{64, 128\}$ *raw* | n/a | n/a |

All methods are given an equal grid search budget of 12 hyperparameter configurations consisting of the two choices of demand input representations, three choices of learning rate $\alpha$, and two choices of model complexity as detailed in Table 2. For the MLP, subsequent hidden layers contain half the units of the first hidden layer. For the GNN-based methods, sum pooling is used to compute a graph-level embedding from the node-level features. Despite potential over-smoothing issues of GNNs in graph classification (e.g., as described in (Chen et al., 2020)), for the flow routing problem, we set the number of layers equal to the diameter so that all traffic entering the network can also exit, including traffic between pairs of points that are the furthest away in the graph.

## C ADDITIONAL RESULTS

Table 3: Properties of the topologies.

| Graph | $N$ | $m$ | Diam. | $\frac{m}{N}$ | Flows in $\mathcal{D}$ |
|---|---|---|---|---|---|
| Aconet | 23 | 62 | 4 | 2.70 | 1587000 |
| Agis | 25 | 60 | 7 | 2.40 | 1875000 |
| Arnes | 34 | 92 | 7 | 2.71 | 3468000 |
| Cernet | 41 | 116 | 5 | 2.83 | 5043000 |
| Cesnet201006 | 52 | 126 | 6 | 2.42 | 8112000 |
| Grnet | 37 | 84 | 8 | 2.27 | 4107000 |
| Iij | 37 | 130 | 5 | 3.51 | 4107000 |
| Internode | 66 | 154 | 6 | 2.33 | 13068000 |
| Janetlense | 20 | 68 | 4 | 3.40 | 1200000 |
| Karen | 25 | 56 | 7 | 2.24 | 1875000 |
| Marnet | 20 | 54 | 3 | 2.70 | 1200000 |
| Niif | 36 | 82 | 7 | 2.28 | 3888000 |
| PionierL3 | 38 | 90 | 10 | 2.37 | 4332000 |
| Sinet | 74 | 152 | 7 | 2.05 | 16428000 |
| SwitchL3 | 42 | 126 | 6 | 3.00 | 5292000 |
| Ulaknet | 82 | 164 | 4 | 2.00 | 20172000 |
| Uninett2011 | 69 | 192 | 9 | 2.78 | 14283000 |

**Topologies used.** High-level statistics about the considered topologies are shown in Table 3.

**Impact of topological characteristics on PEW relative performance.** Figures 6 to 9 compare the percentage changes in NMSE between PEW and the other learning architectures. The results are consistent with the standalone analysis presented in the main text: namely, given the same number of observed traffic matrices, the performance of PEW deteriorates as graph size increases, but improves with higher levels of heterogeneity in node and link-level properties.

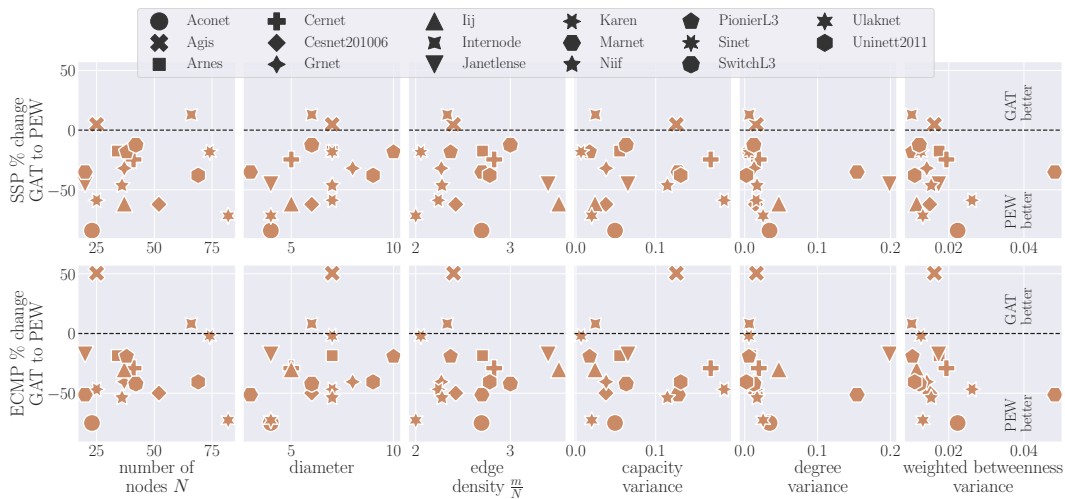

Figure 6: Relationship between the percentage changes in NMSE from GAT to PEW and the topological characteristics of the considered graphs.

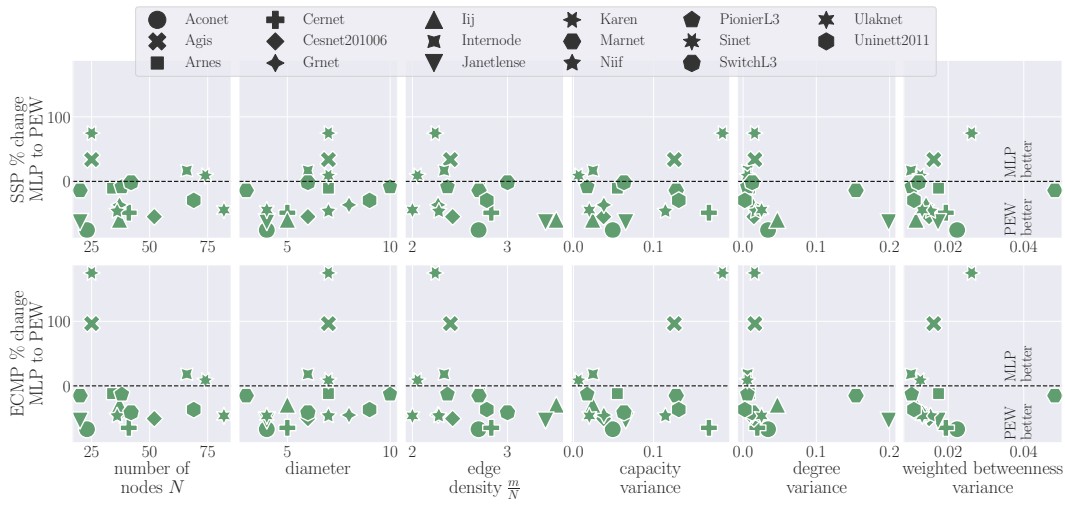

Figure 7: Relationship between the percentage changes in NMSE from MLP to PEW and the topological characteristics of the considered graphs.

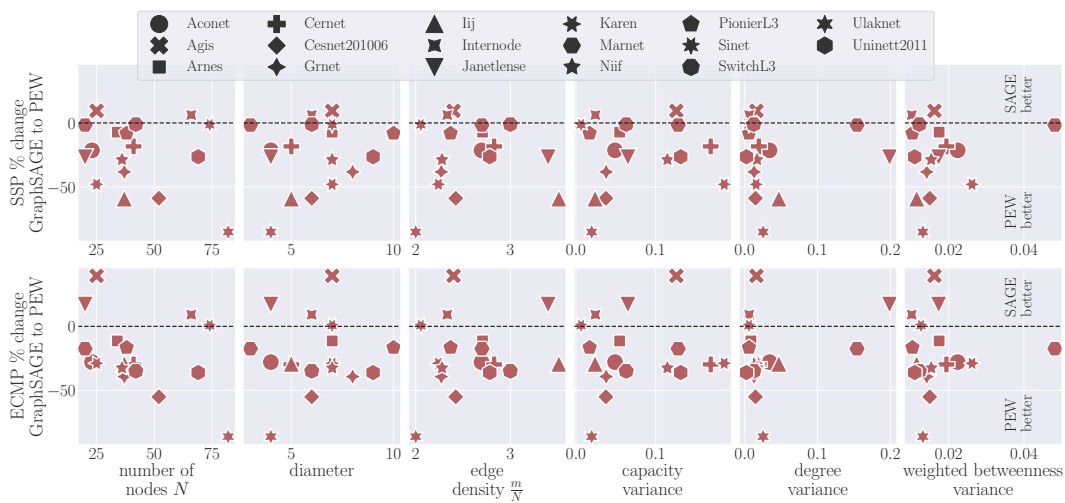

Figure 8: Relationship between the percentage changes in NMSE from GraphSAGE to PEW and the topological characteristics of the considered graphs.

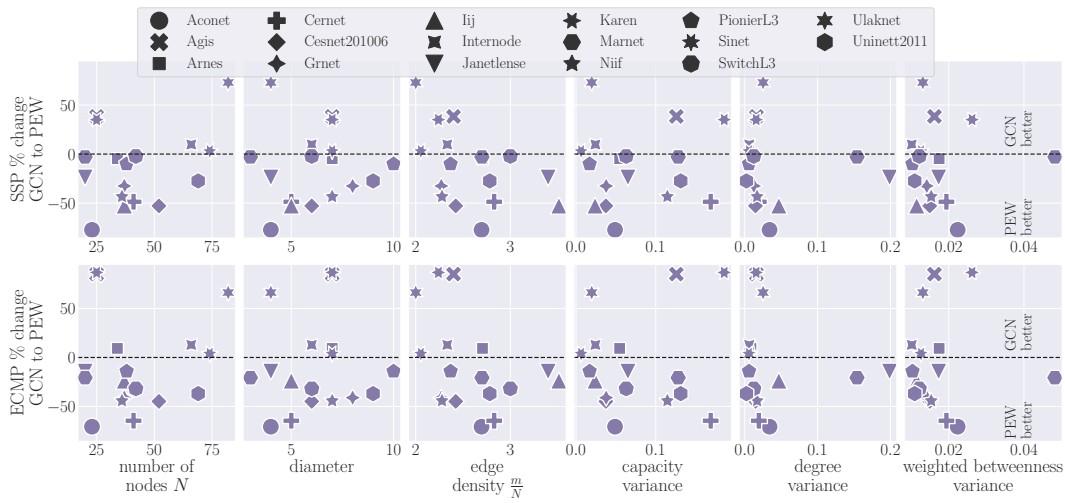

Figure 9: Relationship between the percentage changes in NMSE from GCN to PEW and the topological characteristics of the considered graphs.

**Learning curves.** Representative learning curves are shown in the remainder of this Appendix. For their generation, we report the MSE on the held-out validation set of the best-performing hyperparameter combination for each architecture and demand input representation. To smoothen the curves, we apply exponential weighting with an $\alpha_{\text{EW}} = 0.92$. This value is chosen such that a sufficient amount of noise is removed and the overall trends in validation losses can be observed. We also skip the validation losses for the first $5$ epochs since their values are on a significantly larger scale and would distort the plots. As large spikes sometimes arise, validation losses are truncated to be at most the value obtained after the first $5$ epochs. An interesting trend shown by the learning curves is that the models consistently require more epochs to reach a low validation loss in the ECMP case compared to SSP, reflecting its increased complexity.

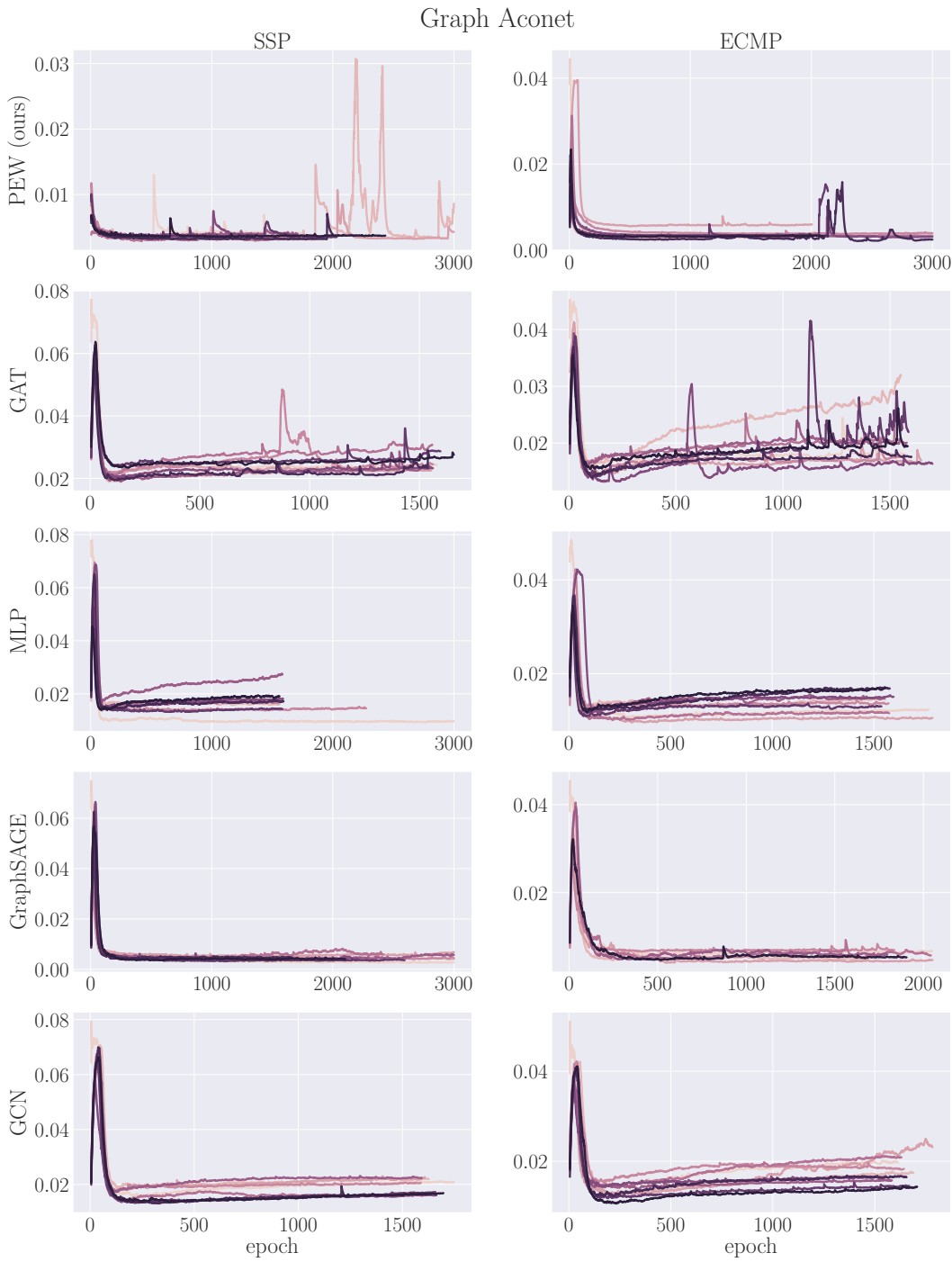

Figure 10: Learning curves for Aconet.

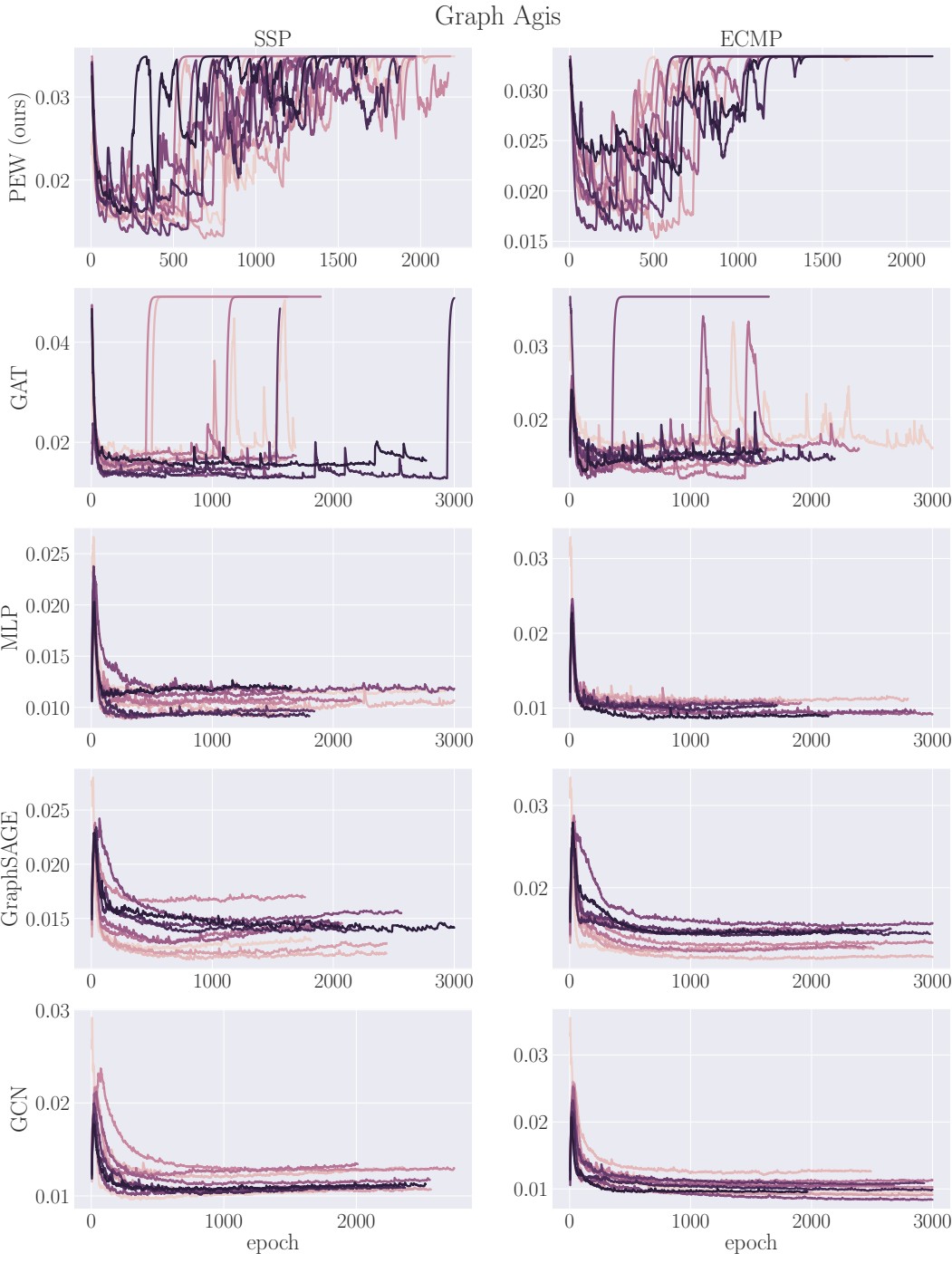

Figure 11: Learning curves for Agis.

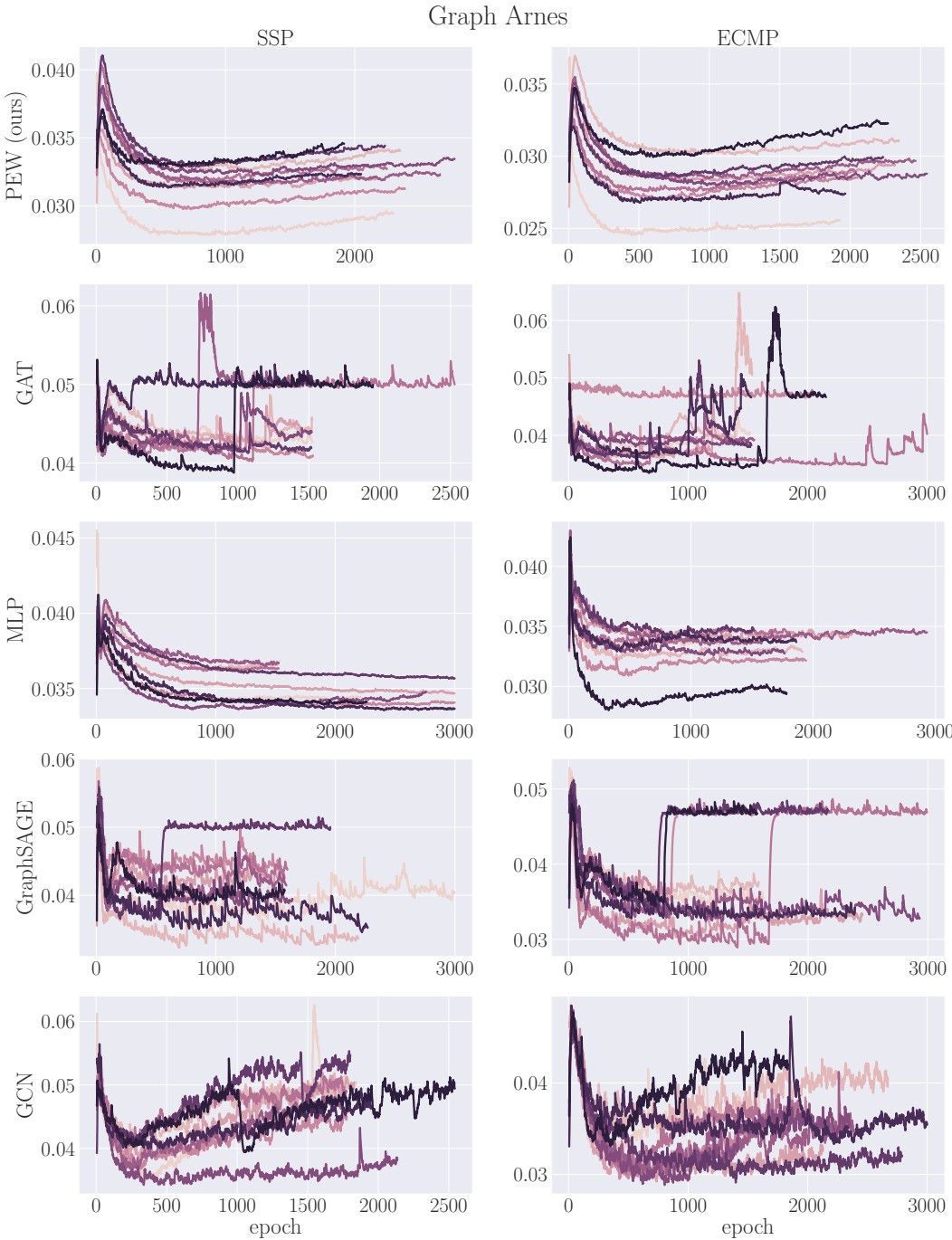

Figure 12: Learning curves for Arnes.

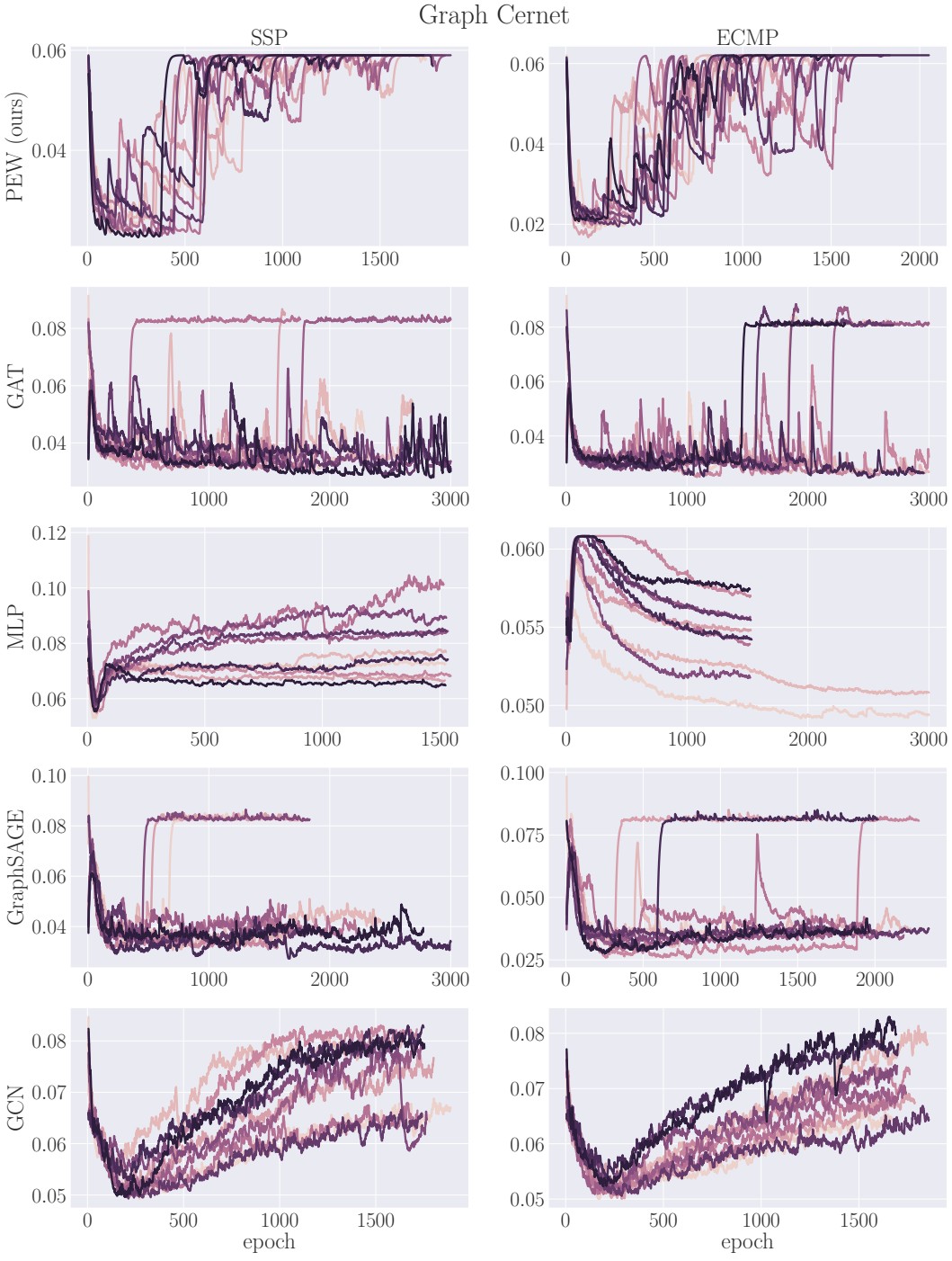

Figure 13: Learning curves for Cernet.

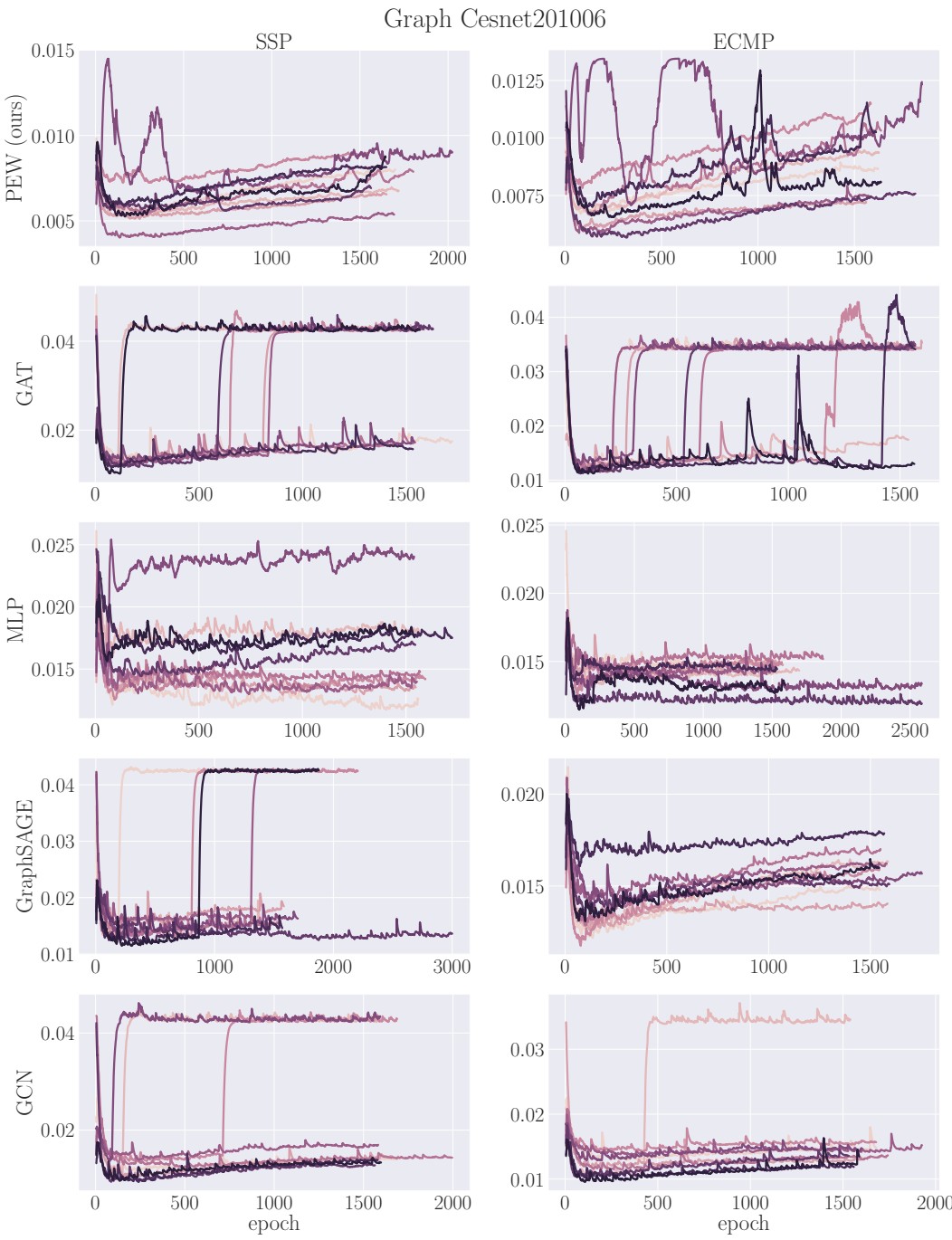

Figure 14: Learning curves for Cesnet201006.

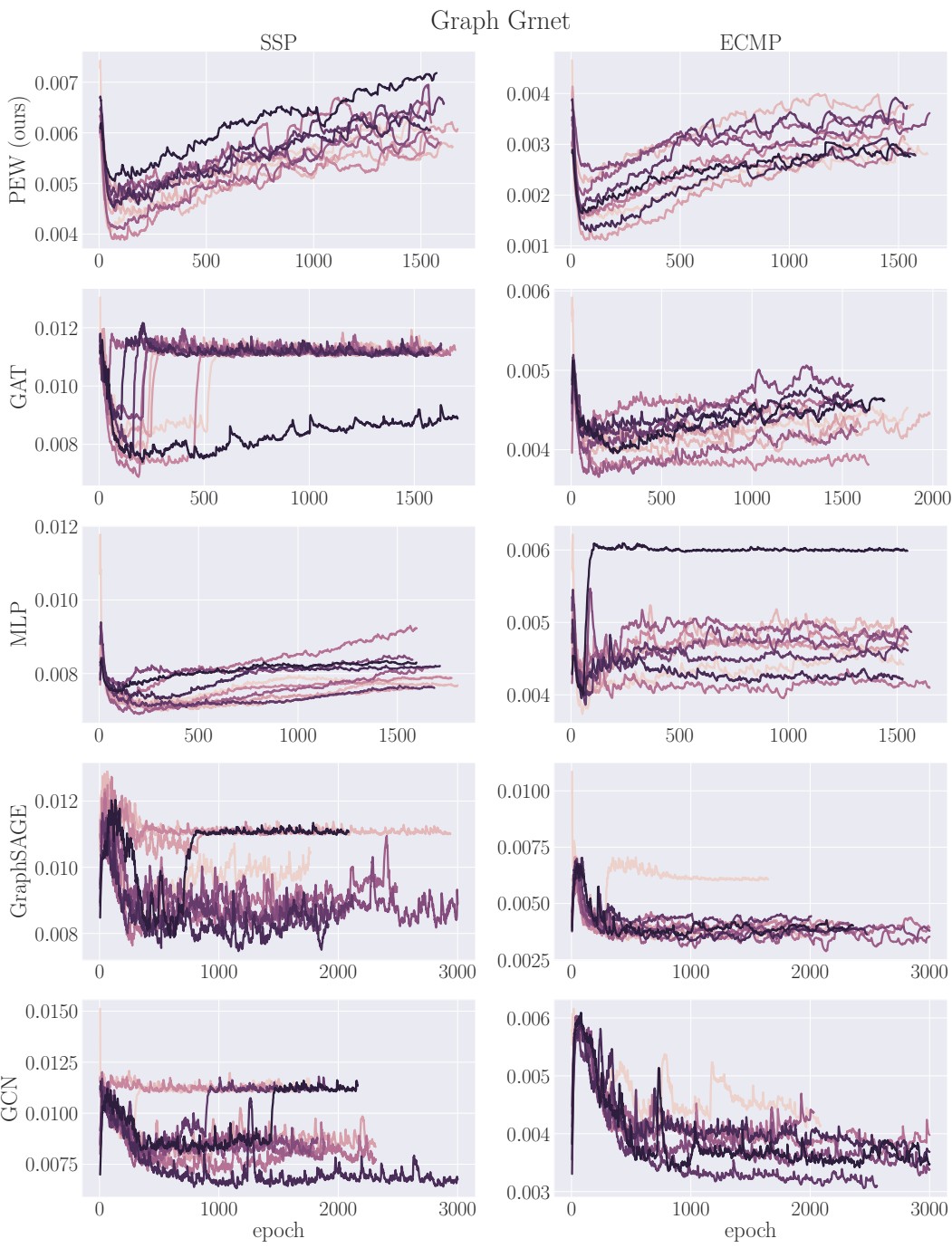

Figure 15: Learning curves for Grnet.

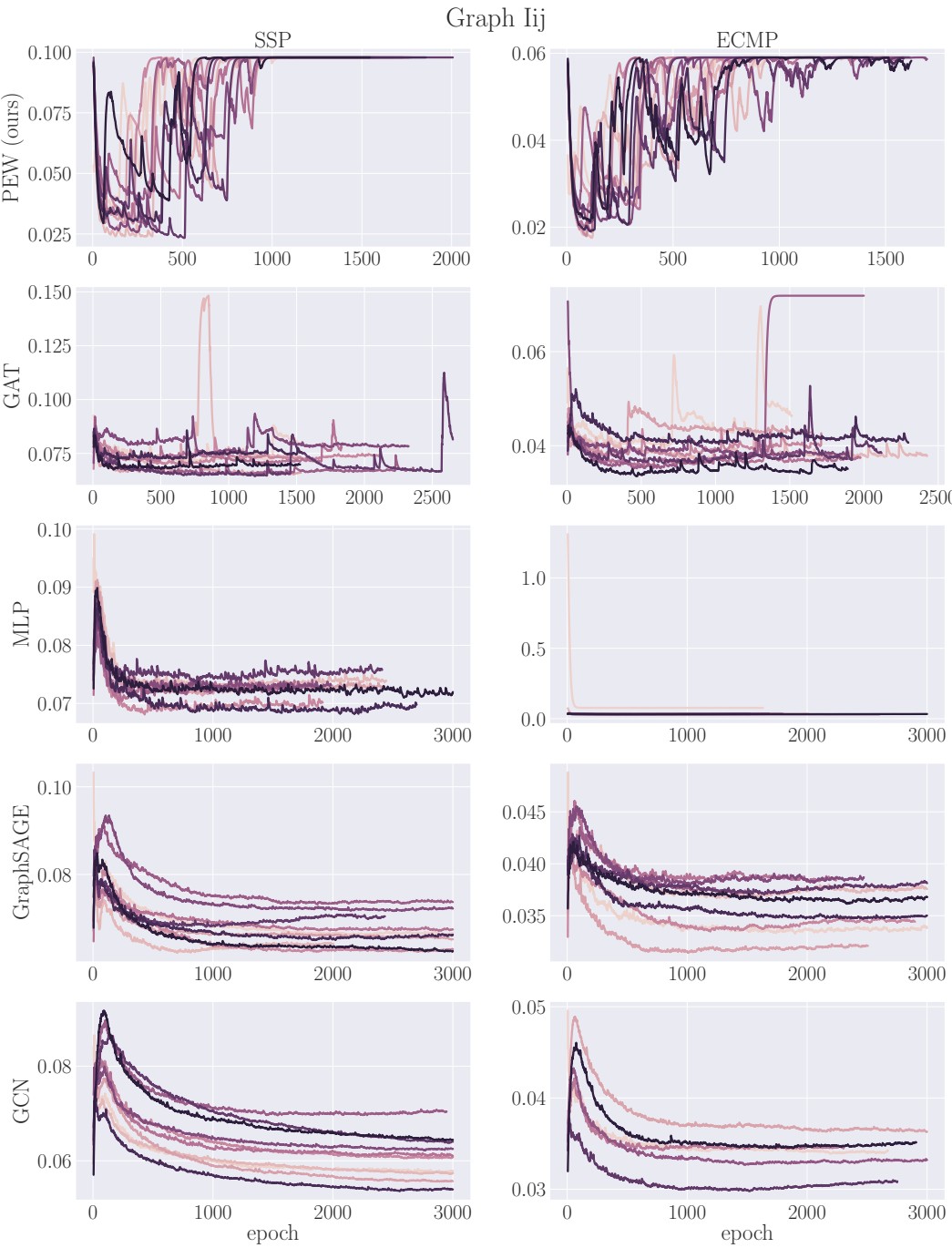

Figure 16: Learning curves for Iij.

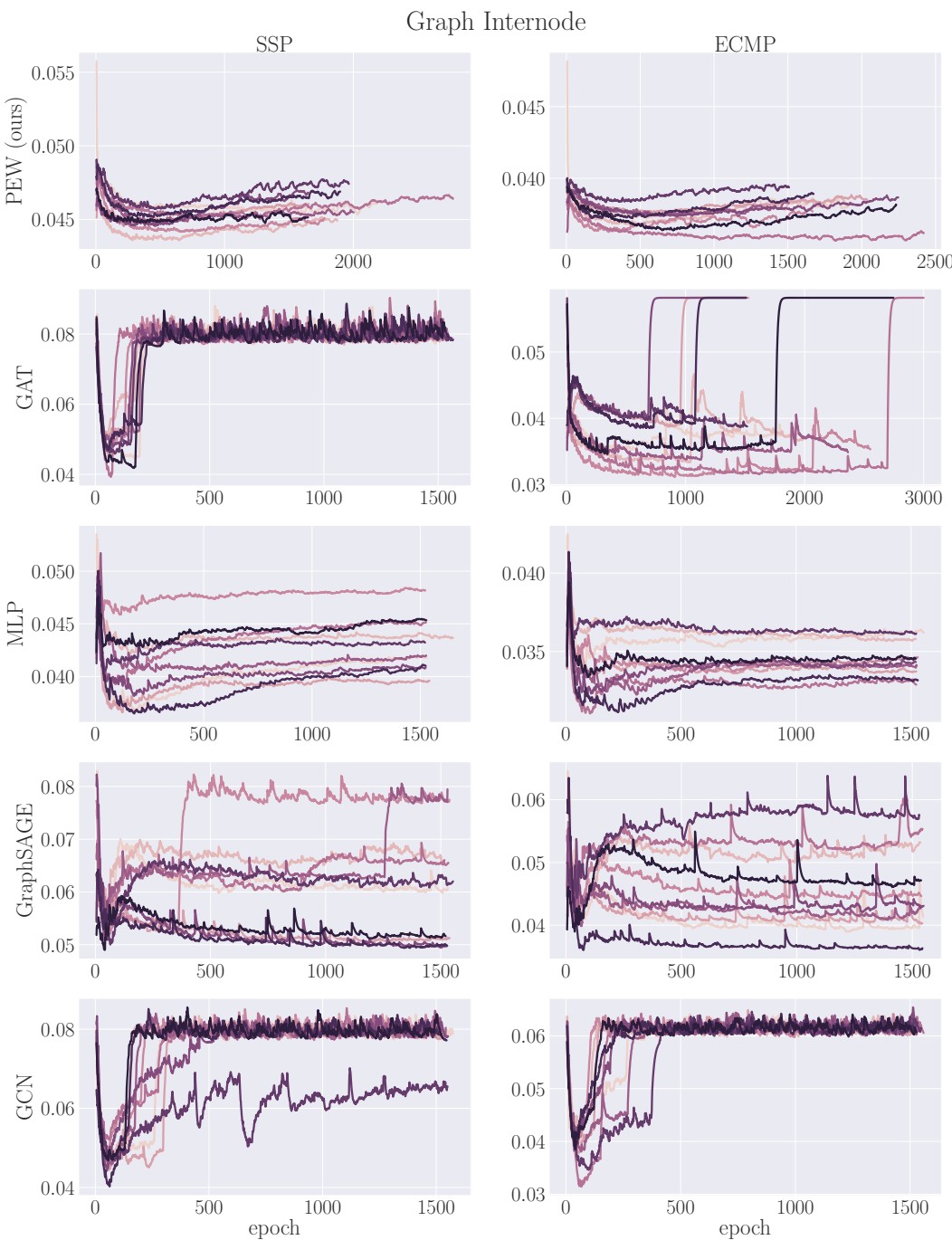

Figure 17: Learning curves for Internode.

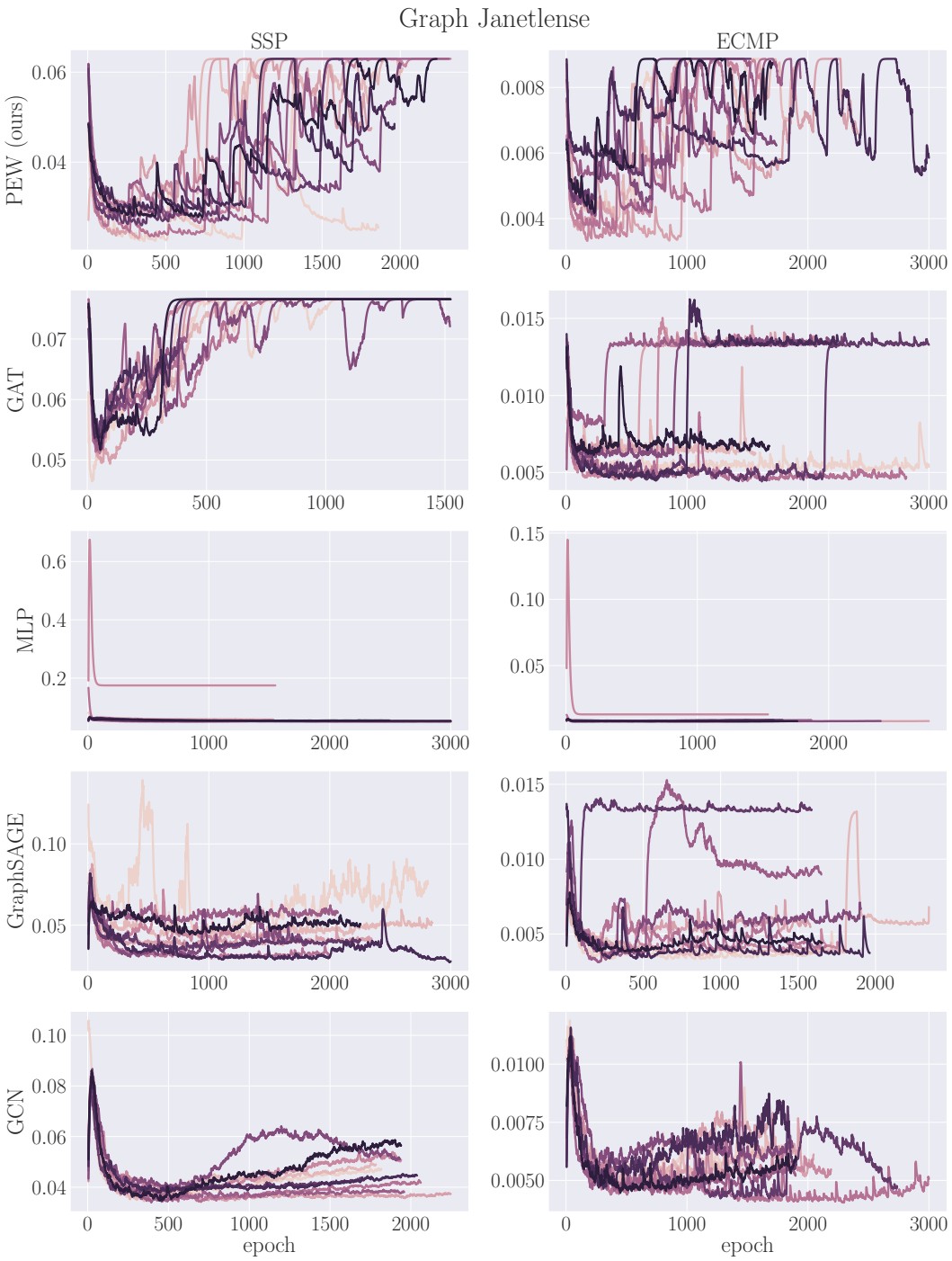

Figure 18: Learning curves for Janetlense.

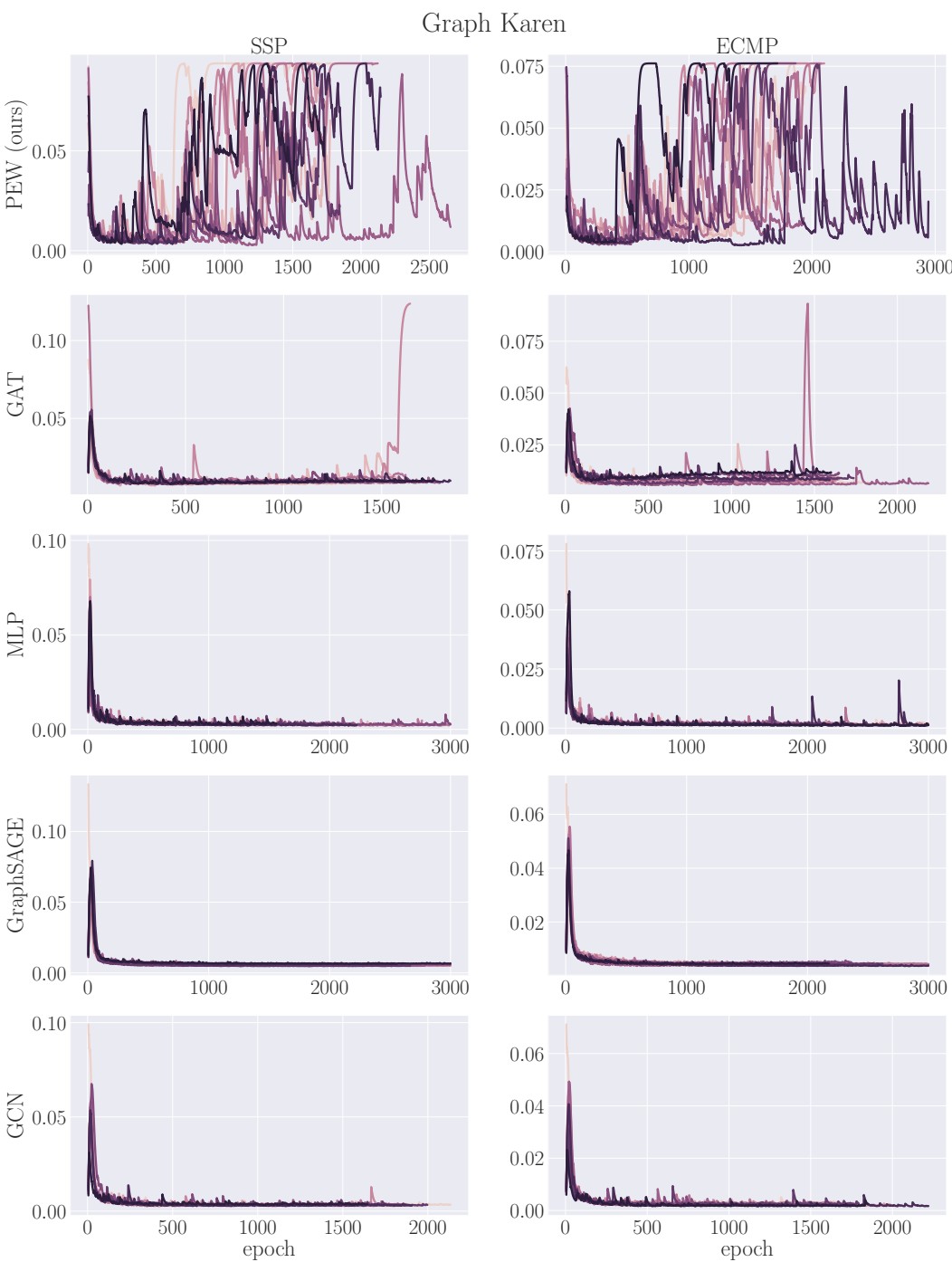

Figure 19: Learning curves for Karen.

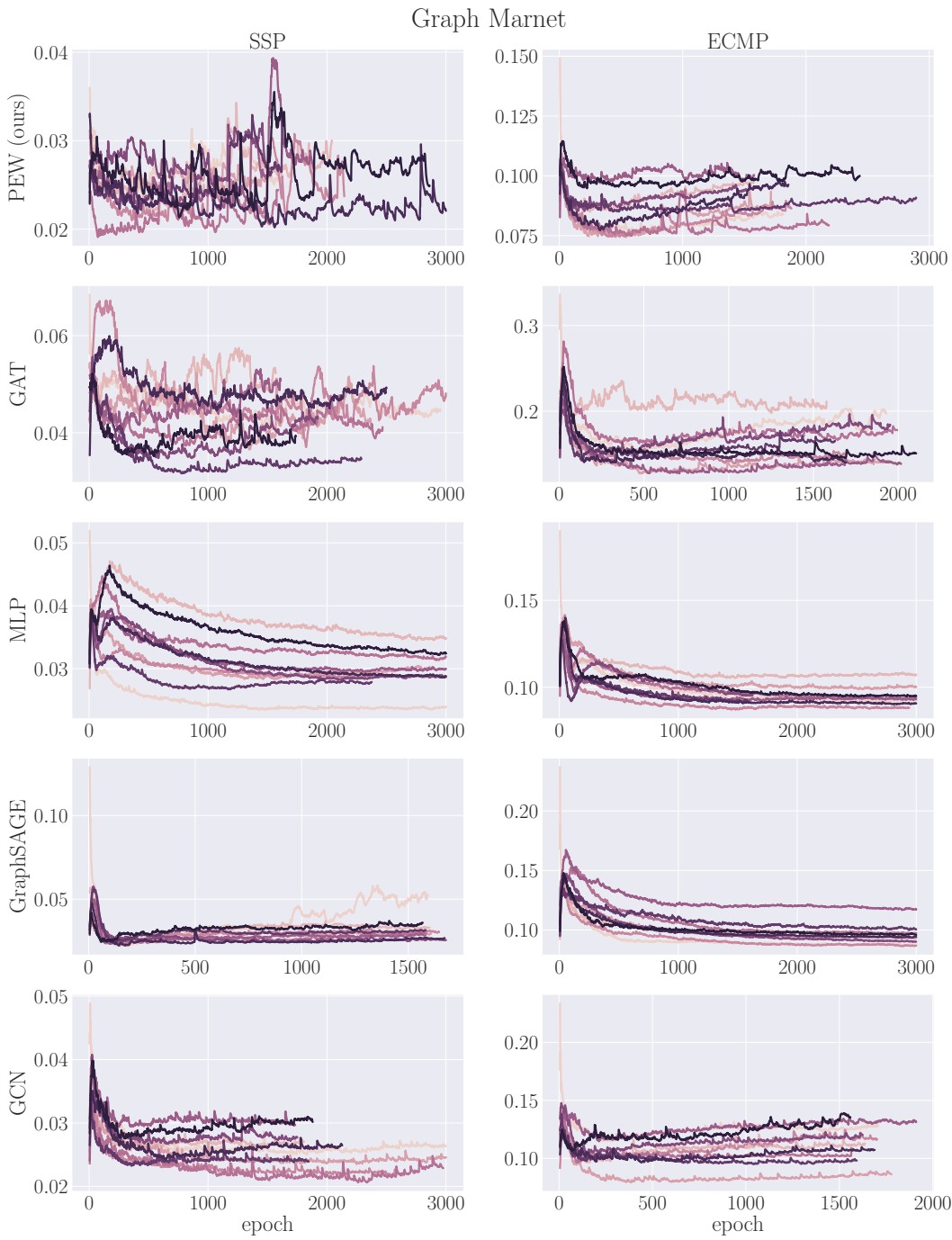

Figure 20: Learning curves for Marnet.

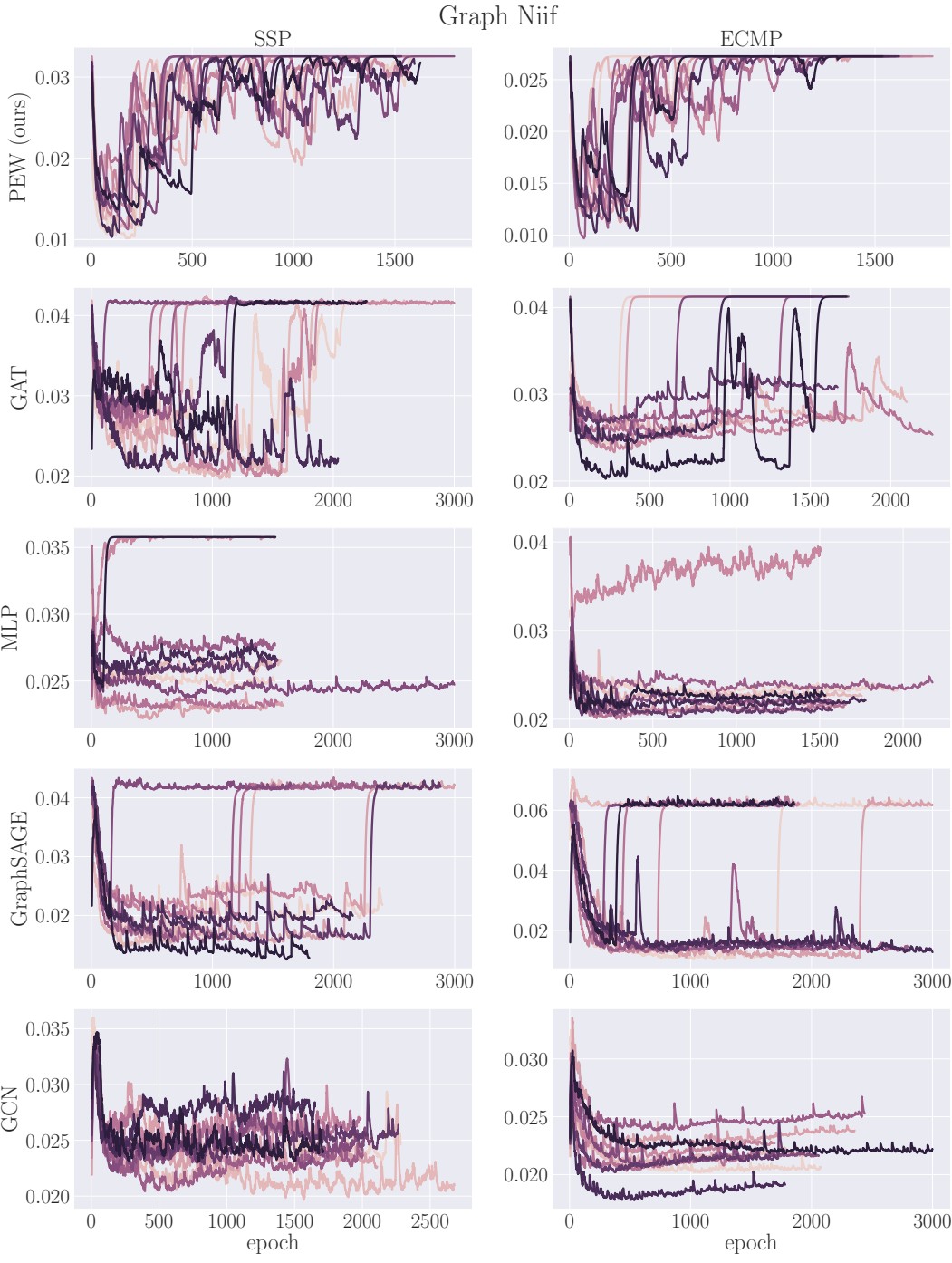

Figure 21: Learning curves for Niif.

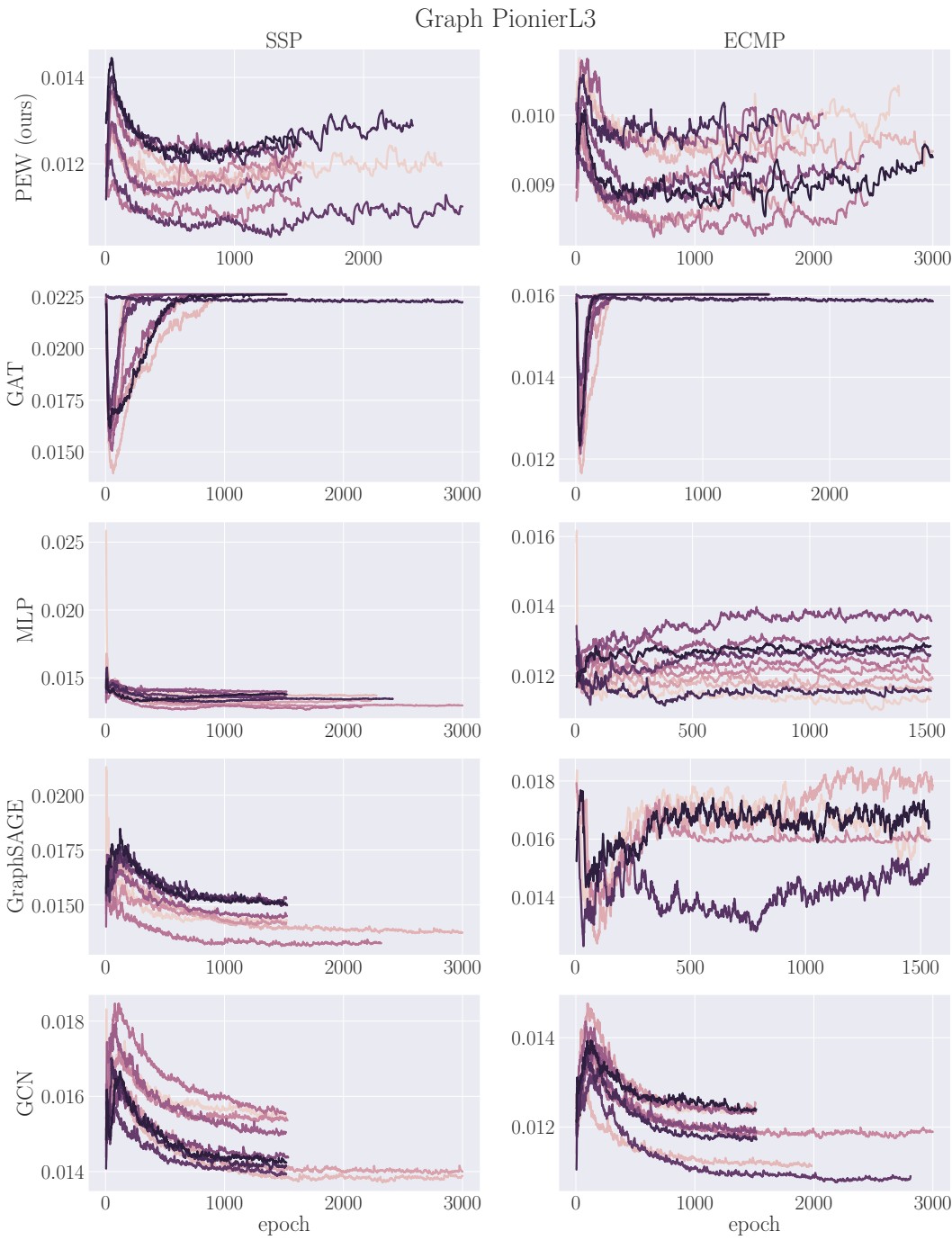

Figure 22: Learning curves for PionierL3.

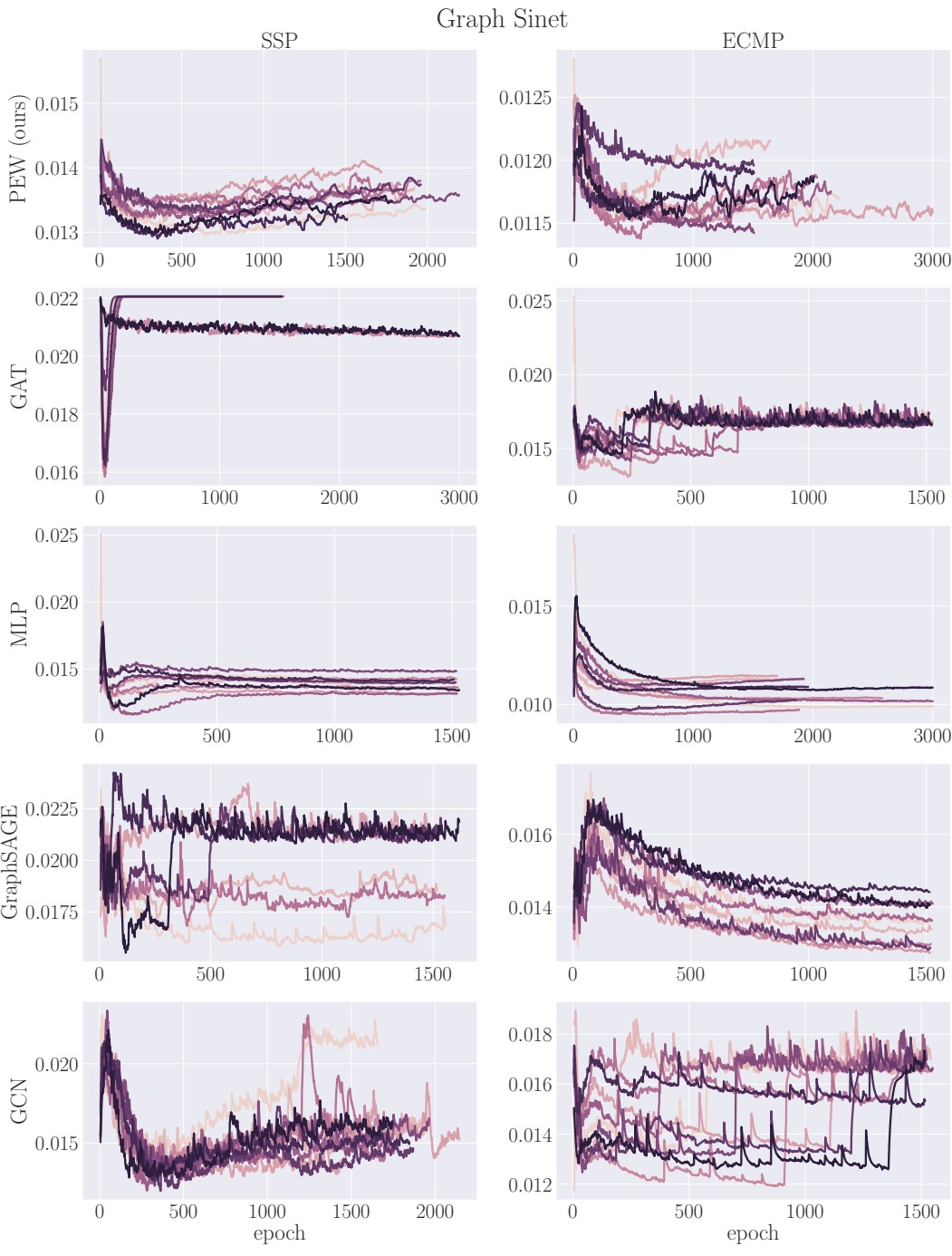

Figure 23: Learning curves for Sinet.

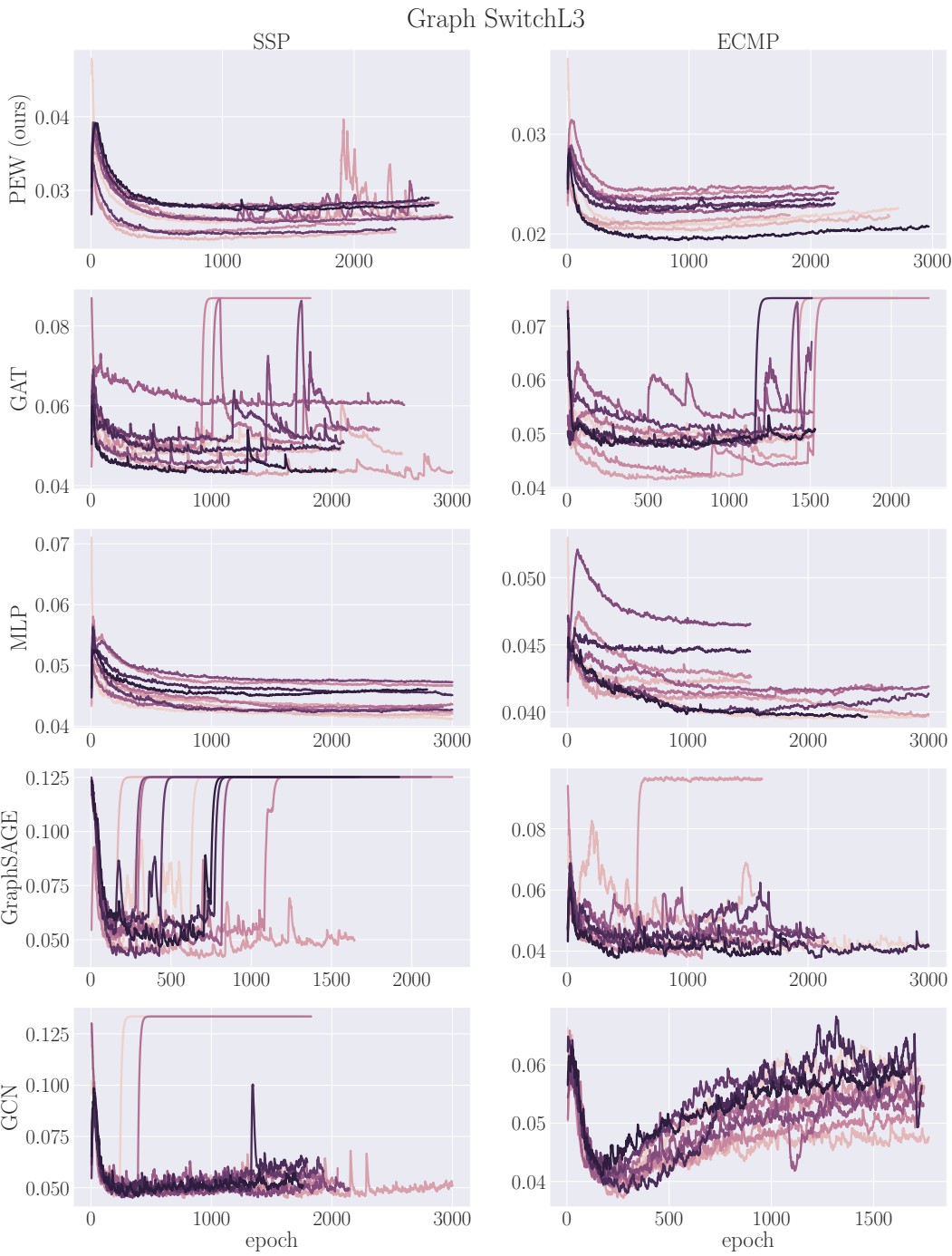

Figure 24: Learning curves for SwitchL3.

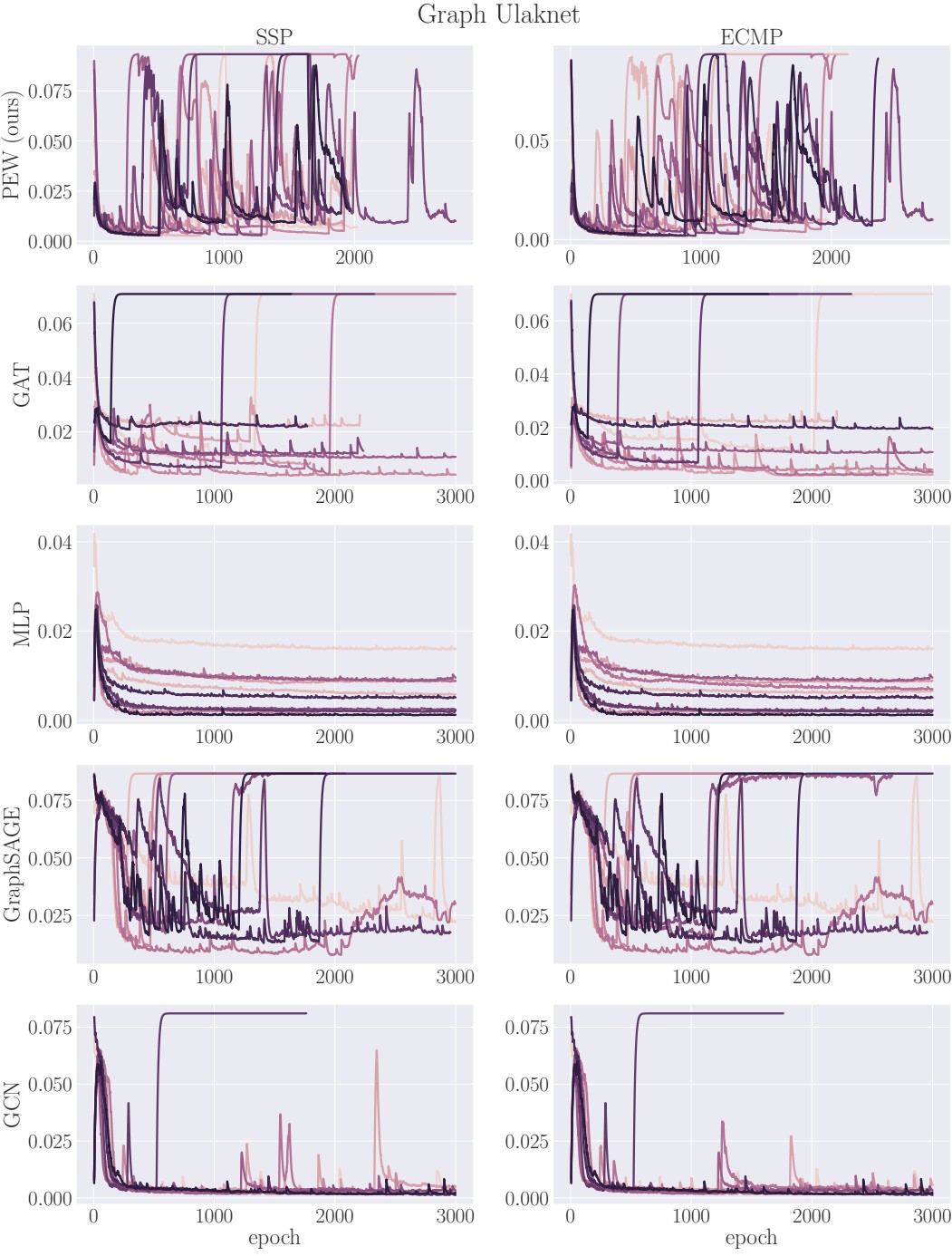

Figure 25: Learning curves for Ulaknet.

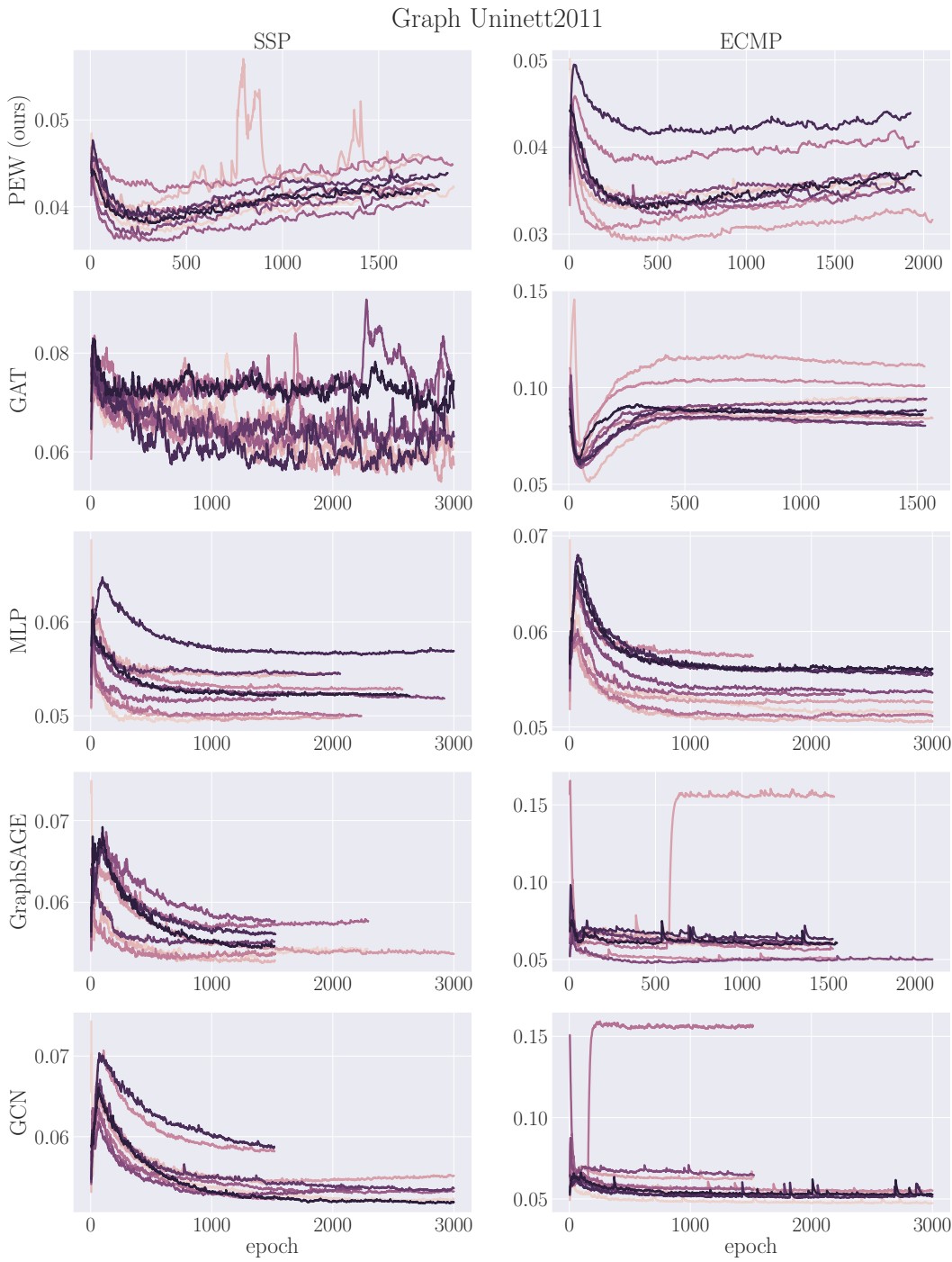

Figure 26: Learning curves for Uninett2011.

