# OpenReview forum: "Graph Neural Modeling of Network Flows"
_ICLR.cc/2024/Conference — Submitted to ICLR 2024_

### Official Review · Reviewer_kHCa · 2023-10-27

**Soundness:** 3 good
**Presentation:** 2 fair
**Contribution:** 2 fair
**Rating:** 3
**Confidence:** 4

**Summary:**

This paper adopts graph neural networks to predict the maximum link utilization of a routing strategy. It presents Per-Edge Weights method where the edges don't have uniform weights. This is different from the traditional message-passing GNNs where the message-passing function is the same for updating all nodes. Empirical study shows that PEW outperforms GAT , GCN, GraphSAGE, and MLP. It is also found that PEW can better utilize the full demand matrix, while GAT can only deal with node-wise sum.

**Strengths:**

1. Extensive empirical studies are adopted to verify the superiority of PEW on the proposed task.

**Weaknesses:**

1. Novelty: The idea of utilizing machine learning methods to solve network flow problems has been around for a few years. Assigning different weights to edges when updating node features is not a completely novel proposal either. Actually, this paper spent only half a page on elaborating its method.
2. The problem of MLU prediction is not as significant as routing design.
3. When identifying the problem, it seems like the graph has no capacity information. When defining MLU, the capacity K is used without notation.
4. Presentation: Too much space is spent discussing the experimental setup. The result presentation is unclear due to the scale of Figure 3 and the vague discussion in Section 5.

**Questions:**

In Section 3.2, the authors argue that the traditional message-passing scheme is not best suited for flow routing problems without demonstrating the reason for this argument.

---

> ### Author Response · Authors · 2023-11-17
> **Author Response to Reviewer kHCa**
>
> We thank the reviewer for taking the time to consider our paper. Please find below our responses to the points that were raised.
>
> > 1. Novelty: The idea of utilizing machine learning methods to solve network flow problems has been around for a few years. Assigning different weights to edges when updating node features is not a completely novel proposal either. Actually, this paper spent only half a page on elaborating its method.
>
> In our opinion, the fact that there have been prior works on a particular problem does not mean that it is not interesting or that it has been solved convincingly. In particular, our work argues that typical learning representations that have been used in past works are not suitable to effectively address the problem. We provide a summary of these works in Section 2.
>
> Our proposed method does not only assign edge weights (as the vanilla GAT), but also considers separate message functions per edges (we would like to stress that this is not the only technical contribution of this manuscript). This is a novel proposal, and we believe that evidence should be provided to the contrary.
>
> We do not see why a simple method and a concise yet clear presentation are reasons for completely dismissing our work, especially given the thorough experimental evidence provided, which the reviewer themselves acknowledges as extensive in their review.
>
> > 2. The problem of MLU prediction is not as significant as routing design.
>
> This is an unsubstantiated and subjective statement that we disagree with – predicting the properties of routing protocols and designing new protocols are both important problems in networking. Both have received extensive attention in the networking literature, as discussed in our paper (see citations of papers from the flagship conferences in computer networking and systems, such as SIGCOMM).
>
> > 3. When identifying the problem, it seems like the graph has no capacity information. When defining MLU, the capacity K is used without notation.
>
> This is not true – notation for capacity is introduced in the “Flow routing formalization” paragraph of Section 3.1 prior to its use in the MLU definition.
>
> > 4. Presentation: Too much space is spent discussing the experimental setup. The result presentation is unclear due to the scale of Figure 3 and the vague discussion in Section 5.
>
> The discussion of the experimental setup (1.25 pages) is required for specifying critical aspects of the results (2.25 pages). Standalone results are meaningless without a precise definition of what is being measured and how. The reviewer should explicitly state what they consider excessive and vague respectively, as we were not able to identify any such parts. The authors do not understand the dismissive tone of this comment, which is subjective in nature in our opinion.
>
> > In Section 3.2, the authors argue that the traditional message-passing scheme is not best suited for flow routing problems without demonstrating the reason for this argument.
>
> This is justified in the paper by mapping the computations performed by message-passing schemes to the computations performed when taking routing decisions, similarly to works that examine the algorithmic alignment of GNNs, as we argue in the introduction. We indeed do not address this formally, since it would require simplifying assumptions on the nature of the message-passing scheme and routing protocol. However, empirical evidence is provided for this claim in terms of extensive experimental results that compare the proposed architecture with its identical message function counterparts. We do agree that this link should be made explicitly and will modify the paper.

---

### Official Review · Reviewer_o9s9 · 2023-10-28

**Soundness:** 3 good
**Presentation:** 4 excellent
**Contribution:** 2 fair
**Rating:** 5
**Confidence:** 3

**Summary:**

This paper presents PEW, a model designed to predict the Maximum Link Utilization, an evaluation metric of routing schemes. The model does not use weight sharing, but learns a different weight matrix for each edge of the network topology. Then, it uses an attention mechanism to compute a weight for each neighbor of a given node. The proposed model is evaluated on a large number of topologies and it outperforms the baselines on most of these topologies.

**Strengths:**

- The experimental evaluation of the proposed model is thorough and convincing. Several different network topologies are considered and also several demand matrices are constructed which led to a very large number of training runs.

- The proposed PEW approach outperforms the baseline models on most datasets. On some datasets the difference in performance between PEW and the baselines is significant.

- The presentation is clear and the paper is easy to read.

**Weaknesses:**

- The proposed method assumes that there is a single network topology which is static and does not change. This suggests that a different model needs to be trained on each network topology and also that a model trained on one topology cannot generalize to other topologies. This renders the proposed approach impractical for several applications.

- Another weakness of the work (which is also discussed in section 6) is that the number of parameters of the proposed model depends on the number of edges of the graph. In case of network topologies that consist of a very large number of edges, this can lead to very large models which are hard to train and might suffer from poor generalization if not many training samples are available.

- It is not clear whether the comparison against the baselines is fair. Are all models evaluated under the same parameter budget? Based on the values shown in Table 2, I would guess that the answer is no. Furthermore, in the case of GCN and GraphSAGE which do not support edge features, the edge capacities are aggregated and used as node features. This is very likely to have a negative impact on those models' performance. I would suggest the authors replace those two models with other architectures that can handle edge features.

- The paper focuses only on a single property of routing strategies (MLU). It is not thus clear whether the proposed method provides performance improvements in tasks where other properties need to be predicted.

- There are some related works which also use graph neural networks to predict the values of other performance indicators that are not discussed in the paper (see for instance [1] and [2]).

[1] Rusek, K., Suárez-Varela, J., Almasan, P., Barlet-Ros, P., & Cabellos-Aparicio, A. (2020). Routenet: Leveraging graph neural networks for network modeling and optimization in sdn. IEEE Journal on Selected Areas in Communications, 38(10), 2260-2270.\
[2] Ferriol-Galmés, M., Paillisse, J., Suárez-Varela, J., Rusek, K., Xiao, S., Shi, X., Cheng, X., Barlet-Ros, P., & Cabellos-Aparicio, A. (2023). RouteNet-Fermi: Network Modeling With Graph Neural Networks. IEEE/ACM Transactions on Networking.

**Questions:**

- In p.5, the closed and open neighborhood of a node are defined. What is the difference between those two neighborhoods?

- In the appendix, it is mentioned that training and evaluation of the models was performed on CPUs. What was the reason behind that? Wouldn't GPU execution lead to a speed-up?

---

> ### Author Response · Authors · 2023-11-17
> **Author Response to Reviewer o9s9**
>
> We thank the reviewer for taking the time to consider our paper. Please find below our responses to the points that were raised.
>
> > method assumes that there is a single network topology which is static [...]
>
> This is incorrect: we do not assume that the topology is static, and we explicitly include an experiment where we show the model successfully handles different topologies (see the “Varying graph structure” part of the evaluation and Table 1). Indeed, we train models individually for each ISP core network with variations in traffic and structure, which is the natural approach in this setting (as an ISP will be concerned with performance on their network, instead of a model that generalizes across many networks). This individual training is performed by most of the studies on ML applied to routing problems covered in the related work section.
>
> > [...] large models which are hard to train and might suffer from poor generalization if not many training samples are available.
>
> As discussed in Section 6, the model for the largest graph has approximately 800000 parameters, which is orders of magnitude smaller than modern models routinely used for vision and language tasks. Given the linear growth in parameters with network size, we do not view this as a problem in practice given the typical sizes of ISP networks, which rarely exceed 200 nodes. ISP network repositories may be consulted to verify this – see, e.g., Figures 7-11 of Knight et al. 2011 as cited in our paper. We also already discuss possible means of reducing the parameter count.
>
> Furthermore, the stated relationship of large parameter counts leading to poor generalizability if not many samples are available is definitely not true in the deep learning regime, where the classic bias-variance trade-off does not hold. This is documented by the “double descent” literature, which shows (counterintuitively) that deep models can exhibit good generalization properties despite being trained to overfit a few samples (see, e.g., [1]).
>
> > […] architectures that can handle edge features
>
> We do not see why the inclusion of the capacities as node features would have a negative impact; in the extreme case, they simply may not help performance. We would like to receive clarifications from the reviewer about this point. We included these methods due to their very wide use in the graph learning community together with a suitable adaptation for the task. We welcome concrete suggestions from the reviewer about representative methods that should be included: in particular, the reviewer mentions “other architectures”, but it is difficult to identify what we should add. It is worth noting that the most important comparison point (the GAT baseline which supports edge features and is the closest model to ours) is already included.
>
> > [...] tasks where other properties need to be predicted
>
> In this paper we do not claim that the proposed method provides performance improvements in tasks where other properties need to be predicted.  We indeed focus on a single metric, which was chosen because of its wide adoption and practical relevance, as discussed in Section 3.1. The paper includes a detailed analysis of the performance of the proposed solution considering several axes of variation, showing that the findings are robust (routing scheme, considered network, topology variations, demand input representation) for this metric. The analysis of different properties will change the scope of the work.
>
> We agree that the key claims of the paper should specifically apply to this metric. We have checked that this is the case. We do not consider, however, that this represents a weakness of the work itself.
>
> > related works which also use graph neural networks [...]
>
> Thanks for the pointers – we were already familiar with some of the work by the senior authors (see references [2, 38] of our submission). The references suggested by the reviewer, in particular, are follow-up works to [38]. We will add these references to the manuscript.
>
> > In p.5, the closed and open neighborhood of a node are defined. What is the difference between those two neighborhoods?
>
> The closed neighborhood includes the node itself (analogously to how closed intervals include their endpoints), whereas the open one does not. We will clarify this, thanks.
>
> > [...] Wouldn't GPU execution lead to a speed-up?
>
> Executing the training on GPUs would indeed lead to a speed-up for a single task. However, given the computational resources at our disposal and the trivial parallelizability of the ~80000 model training runs, executing them on many CPU cores at once yields a shorter total wall clock time. We included this information for reproducibility so that the computational cost of our experiments is specified.
>
> [1] Belkin, M., Hsu, D., Ma, S., & Mandal, S. (2019). Reconciling modern machine-learning practice and the classical bias–variance trade-off. Proceedings of the National Academy of Sciences, 116(32), 15849-15854.

---

> > ### Comment · Reviewer_o9s9 · 2023-11-21
> >
> > I thank the authors for their response. After reading the response, I think most of my concerns are addressed and I have raised my score.
> >
> > I am still not sure whether the comparison against the baselines is fair. As I mentioned in the review, there exist several models that can work with edge features. For example, in [1] all the model names that are followed by the characters "-E" (such as GatedGCN-E and 3WLGNN-E) denote models that take edge features into account, and in several cases, edge features lead to significant performance improvements. Such models are also implemented in the [PyTorch Geometric package](https://pytorch-geometric.readthedocs.io/en/latest/generated/torch_geometric.nn.conv.GINEConv.html). Thus, I would suggest the authors compare their model against such type of architectures.
> >
> > I also feel that the contribution is a bit limited since the paper focuses on a single property. The topic and findings would be of interest to more individuals in the graph machine learning community if the paper considered other properties as well.
> >
> > [1] Dwivedi, V. P., Joshi, C. K., Luu, A. T., Laurent, T., Bengio, Y., & Bresson, X. (2023). Benchmarking Graph Neural Networks. Journal of Machine Learning Research, 24(43), 1-48.

---

### Official Review · Reviewer_ci3b · 2023-10-30

**Soundness:** 2 fair
**Presentation:** 2 fair
**Contribution:** 2 fair
**Rating:** 3
**Confidence:** 4

**Summary:**

This paper proposes a graph neural network architecture called Per-Edge Weights (PEW) for predicting network flows in multi-commodity network flow (MCNF) problems. The key contributions are:

- PEW uses distinct parametrizations to enhance expressiveness when aggregating messages from neighboring nodes along each edge.

- The paper conducts an extensive evaluation on 17 real-world network topologies and 2 routing schemes, totaling over 80,000 experiments.

- The results show PEW improves over standard GNN architectures for predicting maximum link utilization. A well-tuned MLP is competitive with other GNNs.

- The paper analyzes how topological characteristics relate to model performance. Performance tends to decrease with larger graph size but improve with greater heterogeneity in node/edge properties.

**Strengths:**

The paper proposes a novel and simple modification to graph neural networks for network flow prediction problems by using per-edge weight matrices. The writing is mostly clear and easy to follow. Extensive experiments are presented. The analyses provide valuable insights.

**Weaknesses:**

Regarding PEW:

- Limitation for handing diverse topologies. Graphs are inherently dynamic and can have varying sizes and structures, and GNNs are typically designed with the flexibility to handle that. It is unclear to me how can PEW handle graphs with varying sizes and structures. Do we have to train a respective PEW-GNN for diverse real-world graph topologies? Is there a solid reason for just overfitting a specific topology?

- Variance to node permutation. One importance feature a GNN should possess is invariance to node permutation. It is unclear to me if PEW have such a feature, or if PEW doesn’t need this feature.

- Unclear generalizability. PEW seems to be based on some strong assumptions (“node identities are known”) and specifically designed for one problem. What other tasks do you anticipate PEW can excel at?

- Unclear scalability. The handled graphs have only dozens of nodes. PEW wouldn't lead to an unreasonable number of parameters for such graph scales. However, is this a reasonable scale for real-world applications?


Regarding experimentation:

- Potentially unreasonable baselines. Prior works related to "ML for routing flows in computer networks" are summarized in the "Related Work" section. However, both discussions of and comparisons against these works are absent. Instead, the authors compare PEW to some widely applicable standard GNNs, which is not convincing enough for me.

- Absence of comparisons with non-learning baselines. Are there any non-learning baselines for this task? If so, comparing PEW against them would be more convincing.

- Insufficient validation of motivation. How are the ground truth labels generated? What are the advantages of learning-based prediction over the way the ground truth is generated? Are they verified in the experiments?

**Questions:**

Please refer to the weaknesses.

---

> ### Author Response · Authors · 2023-11-17
> **Author Response to Reviewer ci3b**
>
> We thank the reviewer for taking the time to consider our paper. Please find below our responses to the points that were raised.
>
> > Limitation for handing diverse topologies [...]
>
> Our method can successfully handle topologies with varying sets of nodes and edges -- we explicitly include an experiment where we consider this case (see the “Varying graph structure” part of the evaluation and Table 1). If a node or edge are not present, or new ones become active, this is transparently handled by the message-passing procedure.
>
> Indeed, we train models individually for each ISP core network with variations in traffic and structure, which is the sensible approach in this setting (as an ISP will be concerned with performance on their network, instead of a model that generalizes across many networks). This individual training is performed in most studies about ML for routing covered in the related work section.
>
> > Variance to node permutation [...]
>
> Great question. PEW does exhibit invariance to node permutations, since applying an arbitrary permutation to the node order does not change the output of the model. We make the following additional observations: the permutation also needs to be applied to the demand matrix to preserve permutation invariance. The corresponding mapping for determining the unique parametrization of the edges trivially follows from the node permutation, i.e., if node A is mapped to X, and B is mapped to Y, edge A->B is mapped to X->Y, and the property is preserved.
>
> > [...] specifically designed for one problem. What other tasks do you anticipate PEW can excel at?
>
> We respectfully disagree that the model is specifically designed for one problem – rather, it is designed for all problems that can be formulated as (Multi-Commodity) Network Flow. MCNF problems appear in scenarios that are unrelated to computer networks, i.e., in logistics and transportation networks, as discussed in the paper’s introduction. We hence expect the method to perform well where this mathematical formalism applies. MCNF also encompasses the widely studied maximum flow problems, which have a single pairwise demand.
>
> Future work might consider evaluations on other topologies, routing protocols, and performance metrics of interest. More broadly, we think the architecture would perform well in situations in which different edge semantics are present. This is the case in, e.g., the relational GNN literature, discussed in the related work section. However, as it is possible to observe from the presentation of the results in this work, we want to be cautious about this – we believe that extensive experimental evidence is needed.
>
> > Potentially unreasonable baselines [...]
>
> Discussions of other related works are included, but none of the methods are strictly applicable to the scenario we consider, as they all study slightly different end tasks using mostly vanilla learning representations. Our discussion focuses on learning representations that are used in these works (primarily MLP and MPNNs with a global message function). We include MLP and several instantiations of MPNNs in our evaluation. We were not able to identify other learning representations that differ meaningfully from the considered baselines, but we welcome suggestions in this sense.
>
> > [...] non-learning baselines [...]
>
> We are not aware of any “strong” non-learning baselines for this task. A trivial non-learning baseline that we do consider (for normalization purposes) is to simply predict the average MLU present in the training set irrespectively of the demand matrix. All learning-based methods exceed this simple baseline, as can be verified in Figure 3 by the fact that all yield a ratio below 1 in mean squared error.
>
> > Insufficient validation of motivation [...]
>
> This is explained in the paper in the ”Traffic generation” paragraph of Section 4. To reiterate, the labels are generated in accordance with the specific routing scheme considered (concretely, SSP and ECMP) from a given traffic matrix and topology. The traffic matrix itself is generated using a realistic simulator that appeared in one of the flagship computer networking conferences. To provide further details about the MLU calculation, first the shortest paths are determined for each source-destination pair. Then, for each pair, the loads of the links along the (possibly multiple) shortest paths are updated with the fractions of the demands between the two nodes.
>
> As we argue in the introduction of the paper, non-learning-based approaches require a priori knowledge of the demand matrix, which is assumed to stay fixed or otherwise requires a disruptive redeployment of routing strategy under common protocols. This is not a realistic assumption as loads in real systems undergo changes all the time (see Feldman et al. 2001 and Fortz and Thorup 2002 as cited in the paper). ML approaches can enable routing protocols that perform well in a variety of traffic scenarios.

---

### Official Review · Reviewer_fEYx · 2023-11-04

**Soundness:** 2 fair
**Presentation:** 3 good
**Contribution:** 2 fair
**Rating:** 3
**Confidence:** 5

**Summary:**

The paper introduces a new graph architecture for network flow problems, where the goal is to predict the maximum link utilization for a given network topology given some demand matrix. The authors argue that traditional graph learning methods such as variants of message passing neural networks or graph networks are not well-suited for this type of problem, because edges in network flow problems do not have uniform semantics. To address this, the paper proposes a new mechanism, called Per-Edge Weights or PEW, which relies on a different message function per edge when aggregating messages received along each edge. This method builds upon across-relation graph attention networks. The proposed methodology is experimentally assessed on 17 real provided topologies and 2 routing schemes (SSP and ECMP). The authors find that PEW tends to outperform other architectures (GAT, GraphSAGE, GCN), while a standard MLP is competitive with the standard graph architectures.

**Strengths:**

1. The paper is motivated by a real problem in flow networks, which is ubiquitous in communications, transportation and logistics. This potentially makes this work relevant to a wider audience, e.g., the communication networks community.
2. The main observation behind PEW is meaningful. One expects that edges cannot play the same role in the underlying flow network, so that we cannot uniformly aggregate the messages received along each edge. The fact that the proposed methodology uses a different message function per edge thus makes sense in such a context.
3. The paper uses real network topologies, which makes the results more reliable. The traffic matrices are synthetic but use the gravity model, which is considered quite realistic in the traffic engineering literature.
4. The results for MSE indeed show that PEW is indeed able to outperform other graph approaches as well as MLP. This is aligned with the authors' claim that the PEW trick can improve performance in flow networks.

**Weaknesses:**

1. The ML/AI contribution seems to be rather limited. Using a different message function per edge as opposed to identical message functions across all edges is meaningful in the context of flow networks, but the novelty is rather low otherwise. The PEW architecture is a variation of the across-relation variant of relational GATS, so the proposed architecture cannot be viewed as a novel contribution of this work.

2. I am not exactly clear what the motivation behind this paper is. The authors describe the challenge that "a priori knowledge of the full demand matrix is an unrealistic assumption, but ML techniques can address this by learning a model trained on past load that can perform well in a variety of traffic scenarios". However, the real network topologies used in this work are rather small. In that case, it is not clear whether an ML model can provide a substantial benefit. Perhaps one could simply run linear programming algorithms to compute the MLU for a new traffic matrix (if demand changes) or for the new topology (if the network topology suddenly changes)?

3. Continuing point (2), if the authors state that the proposed PEW is the correct architecture for network flow problems, then it would be easier to make such a point if PEW showed potential for devising alternative routing strategies. Currently, the authors have simply used the new architecture with a regression task, i.e., that of computing the MLU. If PEW is the right architecture for general flow problems, then it should show the same potential with other tasks (e.g., classification tasks, or the harder task of learning a policy, for instance via RL). The current setting is too specific and in my view it is not sufficient to show the full potential of PEW.

4. One shortcoming of PEW is that it cannot generalize to new, previously unseen topologies, since the message functions depend on the edge. This can make it cumbersome to use this framework in practice, because one needs to train a separate model for each network topology using lots of different data points. Architectures that would be easily adapted to new networks (e.g., with similar number of nodes/edges but very different topology) would be very helpful.

5. The authors experimented with rather small networks, up to 82 nodes. It might have been useful to consider even larger networks, since small networks may be easier to handle with traditional algorithms. Furthermore, it may have made sense to experiment with synthetic topologies as well, e.g., with networks generated with the BA model, the ER model, or the WS model (all are available in the NetworkX graph analysis package). Strong results on a variety of synthetic topologies (possibly with more nodes) would further strengthen the current claims in the paper. That said, the proposed method may not be easily scalable to large graphs due to having many more learnable parameters (especially, as the number of edges increases).

6. I am not sure I understood Table 1. As mentioned in point (2) above, the authors seem to claim that a motivation of this work is to learn a graph model with high predictive performance under changing topology or demands. To show this claim, the authors would have included the MSE on the varied graph structure. Ideally, we would want the MSE to remain low for a high predictive performance. Instead, the authors mention the WR and the MRR, which is only a relative measure which does not properly reflect the absolute predictive performance. Similarly, in Figure 4 the authors only show how PEW performs with raw demands and GAT with sum demands, but they do not show how PEW performs with sum demands or GAT with raw demands.

7. I am not sure it makes sense to use a fixed number of demand matrices for all topologies. Large topologies would naturally need more training points than small topologies. Normally, we use a dataset that is adequate for learning purposes. The fact that the authors used a fixed number of demand matrices everywhere may be distorting the results, especially for larger graphs.

**Questions:**

1. Can PEW perform well with other network flow tasks, outside regression (classification, RL, etc.)?
2. For the small graphs considered in this work, couldn't we rerun the linear programming algorithm to compute the MLU whenever the demand matrices change or the topology changes (e.g., a link fails)?
3. Have the authors tried to experiment with larger networks? what about synthetic networks?
4. Could the authors provide clarifications on Table1 and Figure 4? For Table 1, in particular, does the MSE remain low when we apply the learned model to the changed demand matrix or the changed topology?
5. Why did the authors use a fixed number of demand matrices?
6. Can the learned models generalize to new topologies? At the very least, do they achieve high predictive performance on changed demands on the same topology (see also question 5)?
7. Is it necessary to include all the learning curves on pages 17-33? Does that serve a specific purpose?

---

> ### Author Response · Authors · 2023-11-17
> **Author Response to Reviewer fEYx**
>
> We thank the reviewer for taking the time to consider our paper. Please find below our responses to the points that were raised.
>
> > 1. Other network flow tasks
>
> We consider that this is outside of the scope of the current paper and a worthwhile direction for future work. We would expect this to be the case given that the evaluation shows robust results across a variety of dimensions (topologies, routing schemes, network structure variations, demand representations).
>
> > 2. Using LP
>
> Indeed, approaches such as LP can be used upon changes in the nature of traffic. However, they imply a disruptive redeployment of routing strategy (see, for example, Fortz and Thorup 2002) under the most common routing protocols that use the source-destination routing paradigm. Segment routing technology, which could carry out this dynamic strategy change without negative impacts, is not yet widely deployed [1].
>
> > 3. Larger and synthetic networks
>
> We have not experimented with larger networks than those presented in the paper, but we dispute the implicit premise that the networks are small. We would like to point out that (1) this is a computationally challenging problem and our work carries out training on substantially larger and greater number of topologies than other works on routing with a comparable formulation; (2) ISP backbone networks rarely exceed 200 nodes – see, e.g., Figures 7-11 of Knight et al. 2011.
>
> Regarding synthetic networks, we opted instead for real-world topologies and a traffic model proven to be realistic. As can be observed in available data repositories, ISP topologies do not exhibit random, preferential attachment, or small-world structure. However, we fully understand this point, and we are happy to include results on synthetic topologies (while mentioning these caveats) in the camera-ready version. We are unable to do so in the short remainder of the response window.
>
> > 4. Clarifications on Table 1 and Figure 4
>
> Regarding Table 1, we presented this information in an aggregated format due to space restrictions. In this setting, the MSE values do degrade across the board, which is to be expected given that all other settings are held equal (including the number of total datapoints). However, they still remain meaningful (i.e., well below the baseline performance of 1), and are comparatively similar across topologies (e.g., the Karen network is still an easier task and Arnes is a more difficult task even with topology variations).
>
> To clarify Figure 4: this visualization is based on four performances: PEW with summed and raw demands; and GAT with summed and raw demands. The line marked PEW represents (MSE obtained by PEW with raw demands – MSE obtained by PEW with summed demands), and equivalently for GAT. Hence, the results the reviewer mentions are already included in this figure.
>
> > 5. Use of fixed number of demand matrices
>
> Using a fixed number of demand matrices is one of the possible choices in terms of experimental design – a varying number is also possible, but would require additional assumptions for determining DM count based on the number of nodes. The first panel of Figure 5 shows that the number of nodes does not definitively determine the performance on the task. Since all methods are granted the same data, we do not consider this to be a methodological issue. Using a fixed number of DMs also maps to the practice of collecting hourly or morning / evening snapshots (e.g., as done in Internet measurement work such as Feldmann et al 2011).
>
> > 6. Generalizing to new topologies
>
> With respect to the question regarding changed demands on the same topology, we would like to point out that this is already the case in the results shown in Figure 3; regarding changed demands and variations in topology, we refer the reviewer to the answer to Q4 above.
>
> We do not treat generalization to completely new topologies, i.e., we train models individually for each ISP core network with variations in traffic and structure, which is the sensible approach in this setting (as an ISP will be concerned with performance on their network, instead of a model that generalizes across many networks). This individual training is performed by most ML for routing studies covered in the related work section.
>
>
> > 7. Learning curves
>
> Learning curves were included for completeness of the results but are not core to the findings of the paper and we do not expect the reader must consult them.  Additionally, as we point out in the appendix, an interesting finding emerging is that models consistently require more epochs to reach a low validation loss in the ECMP case compared to SSP, reflecting its increased complexity.
>
> [1] Ventre, P. L., Salsano, S., Polverini, M., Cianfrani, A., Abdelsalam, A., Filsfils, C., ... & Clad, F. (2020). Segment routing: a comprehensive survey of research activities, standardization efforts, and implementation results. IEEE Communications Surveys & Tutorials, 23(1), 182-221.

---

### Meta-Review · Area_Chair_fvPM · 2023-12-10

**Metareview:**

The reviewers considered the approach to be meaningful, found the application to real graphs to add to the paper's contributions.

On the negative side, they remained concerned on several fronts. As indicated in their responses, they gave several suggestions on how the paper could be improved, but were mostly concerned about the novelty, the importance of the contribution, and how experiments were set up/and how the comparison to competitors were set up.

**Justification For Why Not Higher Score:**

Reviewers remained concerned after rebuttal.

**Justification For Why Not Lower Score:**

N/A.

---

### Decision · Program_Chairs · 2024-01-16

Reject